# Genomic epidemiology of SARS-CoV-2 in a UK university identifies dynamics of transmission

Dinesh Aggarwal [1,2,3,4,128 ✉], Ben Warne [1,3,5,128], Aminu S. Jahun [6], William L. Hamilton [1,3,4], Thomas Fieldman[1,3], Louis du Plessis [7], Verity Hill [8], Beth Blane [1], Emmeline Watkins[9], Elizabeth Wright [9], Grant Hall [6], Catherine Ludden[1,2], Richard Myers[2], Myra Hosmillo[3,6], Yasmin Chaudhry [6], Malte L. Pinckert [6], Iliana Georgana[6], Rhys Izuagbe [6], Danielle Leek[1], Olisaeloka Nsonwu [2], Gareth J. Hughes [2], Simon Packer [2], Andrew J. Page [10], Marina Metaxaki [1], Stewart Fuller[1], Gillian Weale [11], Jon Holgate[12], Christopher A. Brown [13,14], The Cambridge Covid-19 testing Centre*, University of Cambridge Asymptomatic COVID-19 Screening Programme Consortium*, The COVID-19 Genomics UK (COG-UK) Consortium*, Rob Howes[13], Duncan McFarlane[15], Gordon Dougan[1,5], Oliver G. Pybus [7], Daniela De Angelis[2,16], Patrick H. Maxwell[1,3], Sharon J. Peacock [1,3], Michael P. Weekes [3,17,129], Chris Illingworth[16,18,19,129], Ewan M. Harrison [1,2,4,20,129 ✉], Nicholas J. Matheson [1,3,5,21,129 ✉] & Ian G. Goodfellow [6,129 ✉]

Understanding SARS-CoV-2 transmission in higher education settings is important to limit spread between students, and into at-risk populations. In this study, we sequenced 482 SARS-CoV-2 isolates from the University of Cambridge from 5 October to 6 December 2020. We perform a detailed phylogenetic comparison with 972 isolates from the surrounding community, complemented with epidemiological and contact tracing data, to determine transmission dynamics. We observe limited viral introductions into the university; the majority of student cases were linked to a single genetic cluster, likely following social gatherings at a venue outside the university. We identify considerable onward transmission associated with student accommodation and courses; this was effectively contained using local infection control measures and following a national lockdown. Transmission clusters were largely segregated within the university or the community. Our study highlights key determinants of SARS-CoV-2 transmission and effective interventions in a higher education setting that will inform public health policy during pandemics.

---

A full list of author affiliations appears at the end of the paper.

The SARS-CoV-2 pandemic has caused substantial morbidity and mortality globally[1,2]. Universities have been considered conduits for transmission due to extensive social networks of young adults, many of whom live communally, and in-person teaching of large groups[3]. Outbreaks of SARS-CoV-2 have been observed in a number of higher education institutions, but the drivers for transmission in these settings are poorly understood[4]. It is speculated that infection dynamics are dependent on transmission chains involving student courses, residence, study year and social networks[5]. Understanding these dynamics is essential in order to devise effective infection control measures while minimising disruption to teaching, research and the mental health of students and staff[6]. Furthermore, while university students are less likely to develop severe COVID-19 disease, there is concern that university outbreaks could seed infections in more vulnerable populations, including staff, the local community, and upon returning home to older relatives[7]. Identifying possible sources of cross-transmission is therefore vital.

Although SARS-CoV-2 genome sequencing has clear utility to identify virus emergence and cryptic transmission[8,9], no large-scale genomic studies in university settings have been conducted. The United Kingdom has an extensive community genomics surveillance programme through COG-UK[10] which complements traditional contact tracing approaches by providing understanding of circulating viral populations.

We report the results of a genomic epidemiology study of SARS-CoV-2 across a complete term at the University of Cambridge (UoC). Importantly, these findings are from a study period prior to the established circulation of variants of concern and the availability of vaccination, with therefore fewer confounding factors. From 5 October to 6 December 2020, the UoC ran PCR-based symptomatic testing for all staff and students, and offered asymptomatic screening to 15,500 students living in university-managed accommodation. We therefore provide a unique study of SARS-CoV-2 infection that encompasses pre-symptomatic and asymptomatic students[11]. Positive samples from the UoC were sequenced and compared with systematic surveillance SARS-CoV-2 sequences from the local community. The results were analysed in conjunction with epidemiological data derived from the screening programme and national contact tracing. Overall, we describe introductions of SARS-CoV-2 into a higher education setting, the dynamics of transmission both within the university and between the university and the surrounding community, and the impact of local and national measures to control the spread of SARS-CoV-2 infections.

## Results

In total, 972 SARS-CoV-2 cases were identified among university students and staff over the course of term (5 October to 6 December 2020). High-quality genomes were generated from 446/778 (57.3%) positive cases from the university testing programme, from 107/266 (40.2%) cases identified through the Healthcare worker (HCW) screening programme (95 HCWs, 8 students, 4 university staff) and 104 patients identified by hospital testing (71 SARS-CoV-2 positive patients from Cambridge University Hospitals (CUH) and 33 from other medical facilities in Cambridgeshire). A further 797 local cases identified by community testing during the study period were present within the COG-UK dataset, of which 17 were identified as students, 7 as university staff and 26 as HCWs (Fig. 1). Of all identified SARS-CoV-2 cases from Cambridgeshire (university and community) during this period, 8.0% were sequenced (Supplementary Fig. 1).

**SARS-CoV-2 lineages and transmission clusters**. Over the 9-week term, 62 Pango lineages were identified across the university and community (Fig. 2a, c). In the university, 23 Pango lineages were identified, and 438/482 (90.9%) cases were from just 4 lineages (B.1.60.7, B.1.177, B.1.36, B.1.177.16), all of which were detected by the second week of term. Twelve lineages were only observed after the second week of term and accounted for 6.9% cases. By comparison, 57 lineages were identified in the local community over the same 9-week period. Viral genomes containing mutations in the spike protein that have been linked to decreased sensitivity to antibody-mediated immunity or impact viral transmission were observed in the university population: three sequences from the B.1.258 lineage containing the N439K mutation and ΔH69/ΔV70; two cases of B.1.1.7/alpha variant and its associated mutations[12]; and 88 cases of B.1.177 with the A222V mutation[13]. Of these, Pango lineage B.1.1.7 is most reliably associated with increased transmission[14]; both cases of B.1.1.7 were amongst postgraduate students with no epidemiological links, during national lockdown, and failed to transmit further within the university.

In total, 198 putative transmission clusters were defined by CIVET (https://github.com/artic-network/civet). Only 8/36 clusters with university cases contained five or more university members (range 6–337), which together represented 91.3% of all university cases, signifying that the majority of introductions into UoC did not cause ongoing transmission. To further investigate the largest of these, cluster 1 described below, we identified groups of identical samples (0 SNP differences) which produced 19 additional clusters (a total of 34 clusters with >2 university cases) for further analysis.

**Determinants of viral spread across the university**. To determine transmission dynamics following introduction into the university, we performed a detailed investigation of the largest genomic cluster (Cluster 1), which accounted for 337/484 (69.6%) sequenced university cases (Fig. 3). This was widely dispersed across the university by the middle of term, affecting students from 29/31 colleges, 28 undergraduate courses and 208 households in university accommodation alone (Fig. 4).

Cluster 1 was classified as belonging to Pango lineage B.1.160.7. No mutations previously noted to be associated with increased transmissibility were observed in this lineage compared to other genomes in the study. Interrogation of the entire COG-UK dataset of samples from 2020 showed that this lineage was first identified in the UK on 4 October 2020, in Wales, before becoming predominantly sampled in the UoC (Fig. 3b). The B.1.160.7 lineage was not identified in the local community until term week 3 (19–25 October 2020). This was supported by the median estimate of the time to the most common recent ancestor of cluster 1, in comparison to its most closely related cluster from Cambridgeshire community isolates of 165 days (C.I. 127–207) prior to the start of term (6 October 2020). Together, these results suggest the university cases were introduced from outside Cambridgeshire. Additional analysis with A2B-COVID[15], which uses genomic data alongside timing of infection data to evaluate plausibility of transmission between individuals, we showed that these sequences were consistent with a single introduction into the university (Fig. 3c).

National and university contact tracing data were used to identify the initial source of dispersion of this cluster. Ten students from the first two weeks of term reported visiting the same nightclub (venue A). Nine individuals either had an isolate from cluster 1 or (in the event that their sample did not yield a high-quality sequence) were household contacts of an individual

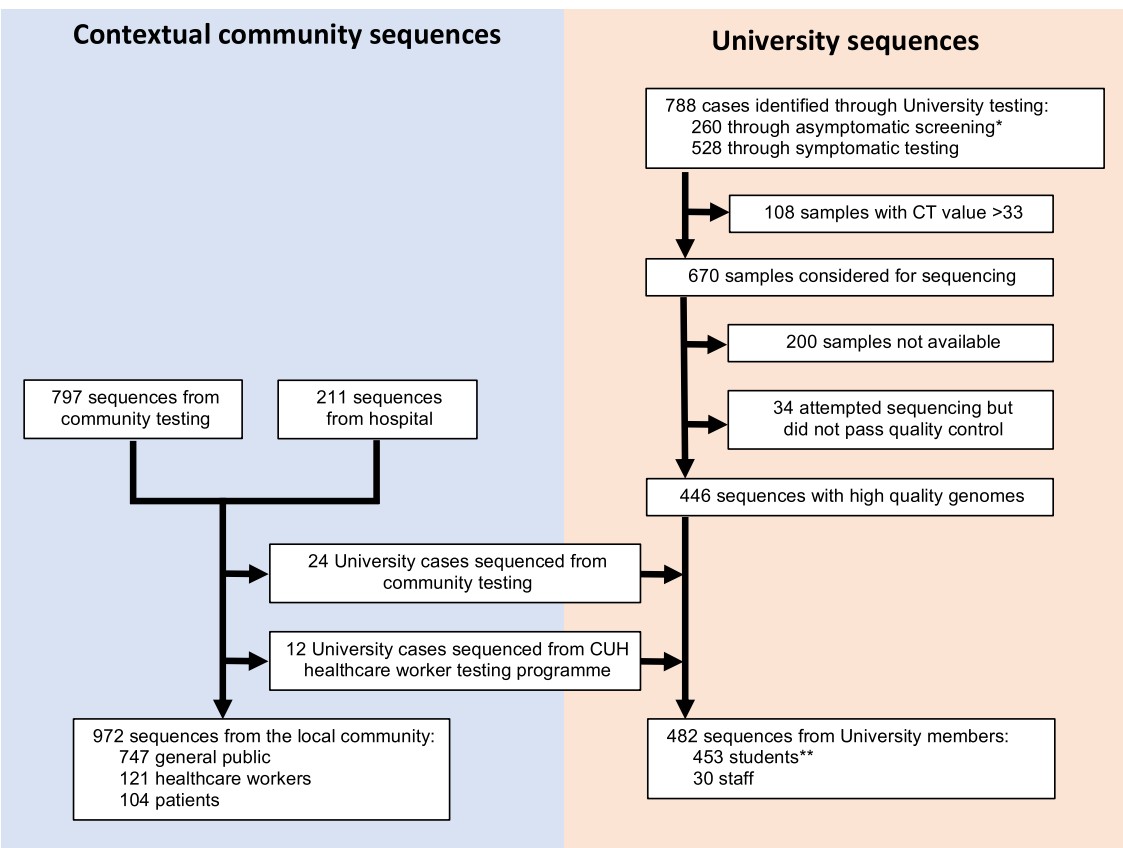

**Fig. 1 Study cohort and available genome sequences.** *Includes 14 students identified through ad hoc asymptomatic screening conducted as part of an outbreak investigation by the University of Cambridge in conjunction with local public health authorities, responding to increased rates of infection in a block of student accommodation (described in further detail in cluster 2 below). **Includes two students associated with a single sequenced pooled sample (see supplementary methods). CUH Cambridge University Hospitals.

with a sequenced cluster 1 isolate. No information was available for one student (Supplementary Fig. 5).

Transmission of cluster 1 was sustained from the first week of term until a national lockdown was enforced on 5th November. Students testing positive in the two weeks around lockdown reported common exposure events predominantly linked to nightclub venues (25/59 (42.4%) of exposures external to the university reported by 48 students). Venue A, identified above as the possible source of dispersion of this cluster at the start of term, was also the most common venue identified in the two weeks around lockdown ($n = 16$). 9/16 cases had sequences in cluster 1, and a further five individuals (where no sequence was available) were household contacts of sequenced cases in cluster 1 (Supplementary Fig. 6).

To determine the impact of lockdown and other control measures within the university, a birth-death skyline model[16] was used to measure changes in the effective reproduction number ($R_e$) within cluster 1. The model indicated an initial $R_e$ at the start of term that was slightly larger than 1, albeit with wide uncertainty (median 1.14; 95% HPD: 0.27–2.21 on 5 October). Over the next 2 weeks $R_e$ continued to rise (median 1.52; 95% HPD 0.94–2.22 on 15 October) followed by a subsequent gradual decline over the next 2 weeks (Fig. 5a). There was a rise immediately prior to the start of lockdown (median 1.55; 95% HPD 1.25–1.86 on 5 November), followed by a steep decrease thereafter (median 0.23; 95% HPD 0.07–0.41 on 19 November) (Fig. 5a), consistent with declining absolute numbers of SARS-CoV-2 infections seen during this time (Fig. 2c). The model estimated the median effective infectious period for individuals in the cluster at 3.03 days (95% HPD:

2.44–3.59 days) (Fig. 5b). As the model does not explicitly incorporate an incubation period and assumes that individuals cannot transmit after being sampled, the effective infectious period represents the mean time from infection until testing positive and assumes perfect infection control measures thereafter. Estimates of $R_e$ and the effective infectious period are robust to model parameterisations (Supplementary Figs. 8–10). Sampling proportion estimates largely overlap with empirical estimates based on the number of positive cases that were sequenced during each week (Fig. 5c). Although sampling proportion estimates are sensitive to the prior specifications, $R_e$ estimates are unaffected (Supplementary Fig. 11).

**Transmission within university households.** There was evidence of transmission of SARS-CoV-2 in student accommodation in 18/34 university clusters. In cluster 1, 169/337 (50.1%) students had a virus genome sequence identical to at least one other student living in the same or neighbouring household (sub-clusters within 0 SNPs ranging between 2 and 11 students).

The largest cluster associated with transmission in accommodation was cluster 2 (lineage B.1.36). By term week 3, this cluster involved 30 students, of which 24 (80%) lived in the same accommodation block in College A and 4 students lived in two separate households in the same college (Supplementary Fig. 12). Interventions from the university, supported by local public health authorities, included isolation of all households in the main accommodation block and individual screening offered to all students. Half of all cases in this cluster were diagnosed by asymptomatic screening. No further genomically-related isolates

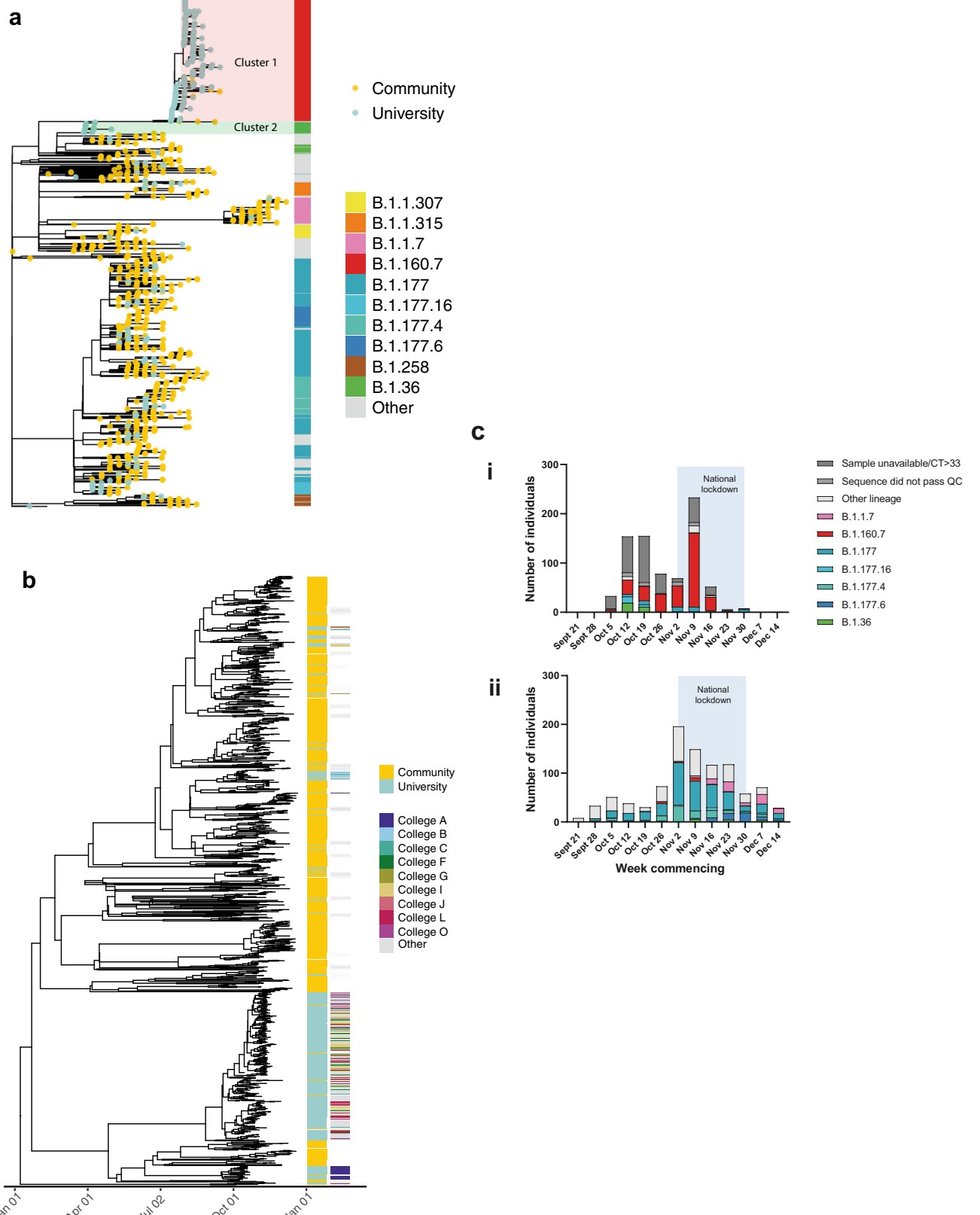

were identified after term-week 3, indicating a successful intervention, and cessation of transmission.

To quantify the importance of household transmission, a Reed-Frost Chain Binomial Model was employed to estimate the household attack rate. Using A2B-COVID[15], we identified 265 households in which the data were consistent with only 1 introduction of SARS-CoV-2. The per household contact probability that an infected person passed on the virus to an uninfected individual within the same household was estimated at 7.8% (95% C.I. 6.9–8.7%).

Further genomic clusters where transmission between house-hold members was implicated are outlined in Supplementary

**Fig. 2 Genomic diversity of SARS-CoV-2 in the university and community. a** Maximum likelihood tree showing that the majority of lineages from university isolates were distinct from community isolates. The node leaves (branch tips) show case location and global PANGO lineage is illustrated in the vertical bar. **b** Time-scaled coalescent tree including university members and local community isolates from study period with visible segregation between the two groups. College affiliation is shown for university members in the second set of vertical columns, highlighting the 'top nine' colleges by cluster 1 prevalence. **c** Epidemic curves demonstrating a steeper decline in SARS-CoV-2 cases in the University of Cambridge (i) compared to the local community (ii), with associated lineages. Only cases with available genomes are included. University term ran from the week commencing October 5 to the week commencing November 30. The light blue shaded area reflects a 4-week national lockdown in the UK, which was associated with a large fall in COVID-19 cases in University students. Specific lineages highlighted are the four largest lineages within the University (minimum 20 cases over the study period) and the community (minimum 50 cases over the study period). For (i), weekly individual case ascertainment for staff and students testing positive for SARS-CoV-2 through both symptomatic and asymptomatic testing pathways provided at the University of Cambridge is indicated. For (ii), weekly cases with genomes available from the local community are shown. Source data are provided as a Source Data file.

Table 1. They follow similar patterns, with groups of cases confined to a single college not leading to sustained transmission.

**Other transmission routes among university members**. In addition to household transmission, there was evidence of viral spread between students in the same course and year of study in 14/34 genomic clusters, with the highest proportion being students in their first year of study. In cluster 1, 203/337 (60.2%) students had an identical isolate to at least one other student studying the same course in the same year (cluster size range 2–14 students). Statistical modelling using data from cluster 1 across the term showed a bias towards infections being observed in first year students ($p$-value = 0.002) (Supplementary Fig. 13, model details in Supplementary Methods). Two further small clusters were comprised of postgraduate students working in the same university department. However, we were not able to determine the probable location of transmission in most cases: there is considerable overlap between course and household clusters, and complex social and study networks exist between students (illustrated in Supplementary Table 1, for example in clusters 3, 4 and 10). Of note, 23/34 clusters with 2 or more genomically linked cases in the dataset contained at least one university member that could not be epidemiologically linked with any other case in their cluster.

The number of SARS-CoV-2 sequences from university staff members were limited in comparison to students ($n = 30$). There was evidence of transmission between staff members working in the same department, college or ancillary role in four genomic clusters. Two clusters contained staff members who shared the same household. There are 8 clusters involving both university staff and students. However, epidemiological associations between these two groups could only be identified in one cluster: a shared household between a student and staff member working in separate university departments.

**Transmission between the university and local community**. We next sought to address the degree of transmission between the university and the local community. Two distinct phylogenetic approaches, shown in Fig. 2, demonstrate segregation of the majority of community and university cases into separate clusters and therefore a lack of substantial cross-transmission. 29/198 (14.6%) transmission clusters contained both university and community cases. Only six clusters contained five or more university cases and included three or more community cases.

To identify transmission clusters involving university and hospital (patient and healthcare worker) cases, we ran CIVET (https://github.com/artic-network/civet) separately with these cases for a focused phylogenetic analysis of this setting. Associations were identified between the university and hospital settings, with 17 clusters involving both university members and either patients or staff. Cluster 1 (69.6% of student cases), contained only 1 patient and 1 healthcare worker with no

identifiable epidemiological link to students. The remaining 16 clusters comprised 133 individuals, including 26 patients, 55 hospital staff or their family members and 52 university members (including 18 staff and 15 clinical medical students). The second-largest cluster of university members ($n = 21$ university and hospital cases) included nine medical students, five healthcare workers and two patients. Phylogenetically, the medical students and one of the healthcare workers were closely linked (Supplementary Fig. 14) and analysis of these cases with A2B-COVID[15] confirmed the plausibility of transmission. All 9 medical students were on clinical rotations at the time of diagnosis of the index case; 7/9 lived in neighbouring households in the same college and the remaining two were named contacts of the index student. Plausible transmission events between this group and the other cluster members were refuted using A2B-COVID (Supplementary Fig. 14).

To further investigate epidemiological associations in clusters involving university members and the local community, 1243/1455 of the cases sequenced over the sampling period were linked to national contact tracing data (excluding hospital cases). 219 (17.6%) cases reported 127 common exposure events. Cluster 1, representing 69.6% of cases within the university, included only 17/976 (1.7%) community cases; only one community case had a common exposure with a university student, dining at the same restaurant. No other epidemiological links were identified in all other genomic clusters. Transmission suspected in 19 epidemiologically linked clusters defined by common exposures was refuted by phylogenetic variation (i.e. identified in separate transmission clusters as defined by CIVET).

**Discussion**

We report the first comprehensive and integrated epidemiological and genomic analysis of SARS-CoV-2 transmission in a higher education setting. Following a limited number of introductions, the majority of cases were linked to a single genetic cluster, that was likely to have dispersed across the university following multiple social gatherings at a nightclub. There was considerable transmission associated with student accommodation and student courses, but minimal evidence of transmission within departments, or between students and staff. We observe the great majority of transmissions occur either within the university or within the local community. Finally, we present evidence demonstrating the efficacy of university measures and national lockdown in reducing COVID-19 cases.

Nearly 70% of all university cases belonged to one genetic cluster (cluster 1), introduced into the UoC by the arrival of students and likely forming a single transmission chain. A nightclub was implicated as an important transmission event at the start of term and again prior to lockdown. This corroborates previous studies identifying such venues as a risk factor for substantial SARS-CoV-2 transmission[17,18]. We urge a cautious approach to the access of such venues during a SARS-CoV-2

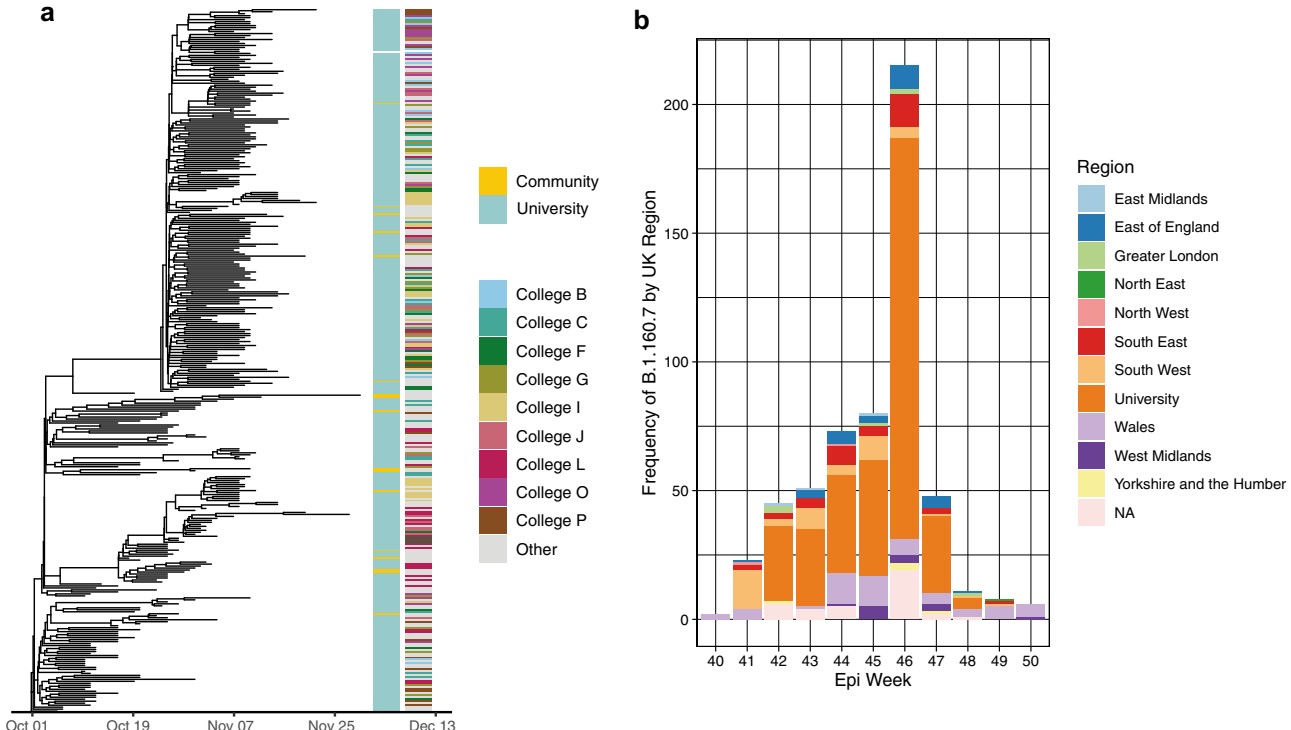

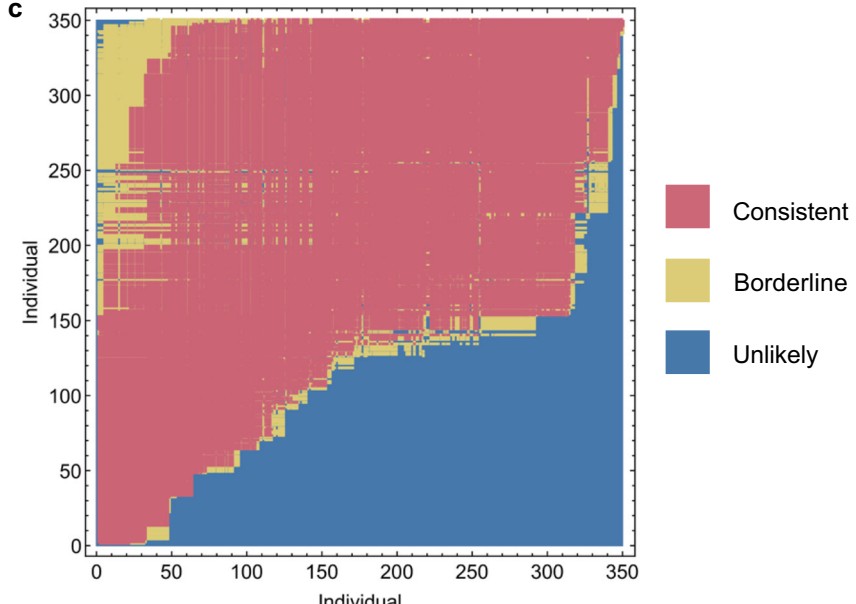

pandemic, particularly in the context of a young susceptible student population.

Our data suggest a substantial change in case numbers and the effective reproduction number over the course of the term. This likely reflects a combination of changes in student behaviour and effective interventions to reduce transmission. Overall, we note that incidence and the effective reproductive number within the university are lower than in other higher education settings and the general UK young adult population during the study period[19]. We highlight a limited number of introductions and low lineage diversity in the university compared to the surrounding community. While the natural extinction of lineages is relatively

common[20], multiple genetically diverse clusters may be expected given the congregation of students from across the globe (international students make up 35% of students in college accommodation)[11]. The lack of diversity may reflect the impact of robust and widely implemented university infection control measures maintained throughout the term, full details of which are provided in the Supplementary Materials, but include social distancing, mask wearing and quarantine of international students at the beginning of term.

There was an initial rise in cases over the first two weeks, coinciding with the first week of term and university Freshers week. This is known to be a period of more intense social mixing

**Fig. 3 Emergence and transmission of SARS-CoV-2 in a large university cluster. a** Time-scaled phylogenetic tree of largest university cluster (cluster 1) derived from the BDSKY model implemented in BEAST 2.6 (Fig. 5). The left-sided heatmap is coloured by case location, and the right-sided heatmap is coloured by student college affiliation, highlighting the top nine colleges by cluster 1 prevalence. Cluster 1 was widely dispersed across the university with limited transmission into the community. **b** Frequency of Lineage B.1.160.7 (to which cluster 1 belongs) in each region of the UK and the University of Cambridge. Regions are defined as 'Nomenclature of territorial units for statistics' (NUTS) regions, where the UK has 9 regions. It is visible that the lineage B.1.160.7 was first sequenced in Wales, and then in the neighbouring South West of England, before becoming prevalent within the University of Cambridge. The lineage remained infrequently detected in the community populating the wider surrounding region (Cambridgeshire, East Anglia, Bedfordshire and Hertfordshire, and Essex, making up East of England) throughout the university term. **c** A continuous transmission chain of SARS-CoV-2 infections in cluster 1 commenced with a single introduction. Relationships between individuals in cluster 1 were calculated within A2B-COVID. Colours denote potential transmission events from the donor (vertical axis) to the recipient (horizontal axis) that are consistent with transmission[12] or which are borderline possibilities (yellow). The plot shows that the data are consistent with a continuous transmission chain of SARS-CoV-2 infections in cluster 1 occurring via a single introduction; there are multiple potential networks of transmission events between these individuals for which each event would be consistent with a statistical model of direct transmission. We note that individuals in this plot are ordered by the date of the first positive COVID test. Source data are provided as a Source Data file.

between students in venues both inside and outside university premises. Between term weeks three and five there was a fall in the effective reproductive number, which coincides with both a reduction in social mixing and the identification of, and subsequent university measures to control, transmission events identified in college residences. In multiple clusters, transmission in student households was successfully interrupted through a combination of measures provided by the university, including rapid case identification through asymptomatic screening, readily available symptomatic testing, contact tracing and comprehensive support provided by colleges for cases and their contacts while in isolation. Further details, including the elaboration of the specific measures to control cluster 2, an outbreak associated with a large accommodation block described above, are provided in the Supplementary Materials. Although we have demonstrated that transmission between students in the same accommodation block is an important factor in the spread of SARS-CoV-2, we report a lower secondary household attack rate (7.8%) than that identified in domestic households (16.6–21.1%) and a lower than expected effective infectious period (3.0 days)[21].

University measures may have been less successful in controlling transmission in settings outside colleges. There was a rise in the effective reproduction number coinciding with the announcement of a national lockdown on 31 October, to begin on 5 November 2020. This announcement prior to implementation of a major socially restrictive public health measure, alongside existing Halloween festivities, may have led to increased levels of behaviour associated with a higher risk of transmission. This supports either reducing the time from announcement to implementation of socially restrictive measures, or the need for a targeted public health campaign to limit high-risk activities where this is not possible. In addition, having identified considerable transmission between students on the same course, we suggest that further mitigation of viral spread may be obtained by implementing shared student accommodation based on university courses.

The national lockdown dramatically reduced case numbers within the university, at a faster rate than the local community, demonstrating high levels of compliance from our study population with an effective control strategy. Contemporary studies conducted elsewhere in the UK have demonstrated that adherence to COVID-19 prevention measures, such as national lockdown, are mixed[22]. Although young age is a risk factor for poor adherence, other associations are less common within the university population, such as having a dependent child in the household, financial hardship and working in a key sector. Although no direct incentives were provided to students, the expectation of individuals to adhere to rules was communicated widely in both national and university media. We also believe that the key to the successful implementation of lockdown was the additional support provided by the collegiate university, ranging from the practical provision of food and drink through to the pastoral and community support provided by established networks of staff, tutors and student representatives.

Finally, we observed limited transmission between the university and the local community. The largest university cluster, accounting for the majority of student infections, was largely phylogenetically distinct from community cases. Further, epidemiological evidence describing common exposures for community and university cases was sparse. However, clinical medical students were disproportionately represented within community clusters. This is an important epidemiological link between secondary care and the university; we highlight this group as being at-risk for both acquisition and transmission of SARS-CoV-2 and medical students should therefore be prioritised for interventions such as vaccination.

A combination of contact tracing and genomics was instrumental to understanding transmission within the university and with its surrounding population; notably in refuting transmission within epidemiologically linked clusters. We advocate for a combined genomic epidemiological approach to inform outbreak investigations as used in other settings[8,23].

This study has a number of limitations. Incomplete sampling and subsequent sequence filtering in both the university and community should be considered when interpreting transmission; the asymptomatic and active case ascertainment in this study should mitigate this discrepancy. The lower community case ascertainment may result in unobserved transmission chains (such as those when assessing the introduction of Pango lineage B.1.160.7 into the university). Further, epidemiological links are dependent on self-reporting and therefore some data will be missing; whilst a lack of epidemiological association between groups in clusters is important and reassuring (such as between staff and students), it does not confirm a lack of transmission. We highlight shared student courses as a risk factor for transmission; this does not take into account the setting of transmission, i.e., during educational or social activities. Finally, the UoC is distinct in its collegiate structure with limited integration with the community; any generalisation of conclusions should be tempered by the study setting.

We present the first comprehensive integrated epidemiological and genomic evaluation of transmission of SARS-CoV-2 within a university. The insights gained will inform public policy regarding infection control measures in higher education settings. We find containment of transmission in student accommodation necessary to mitigate onward propagation. We highlight the

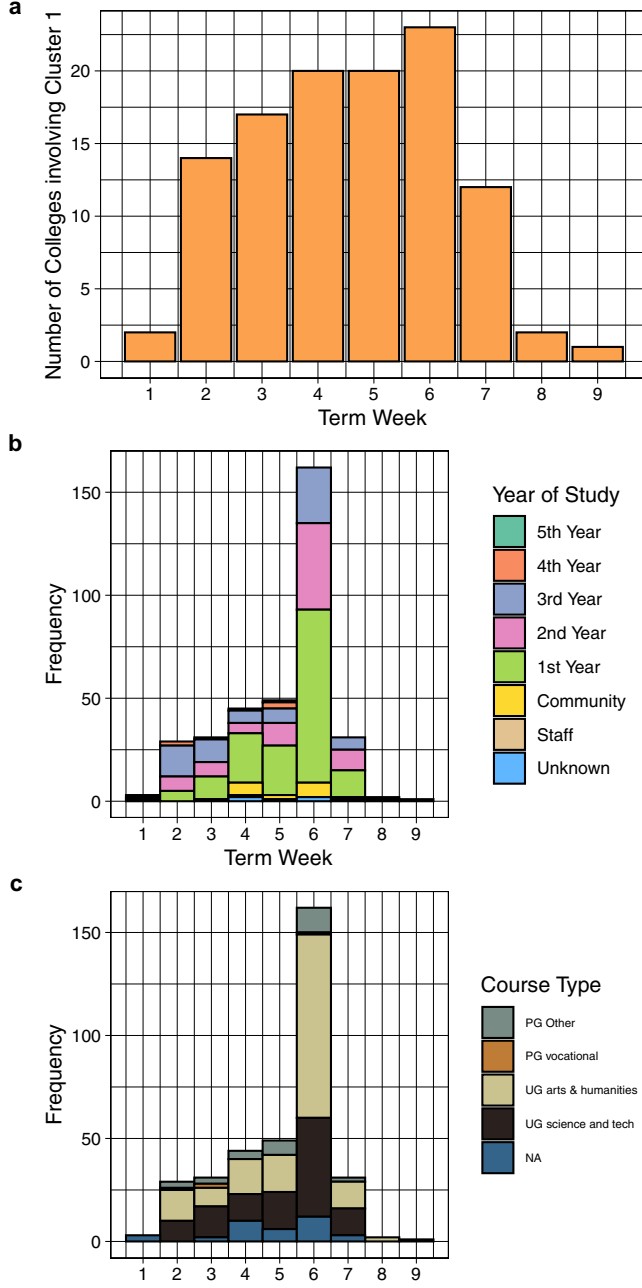

**Fig. 4 Demographics of Cluster 1 across the first university term.**
a Cumulative number of colleges involved in the cluster. Cases included in this cluster were between a number of colleges early during the university term. b Frequency of cases involved in the cluster by year of study.
c Frequency of cases involved in the cluster by course type. Source data are provided as a Source Data file.

importance of targeted public health measures towards nightclub venues to limit transmission. Critically, these findings are likely to be informative for future pandemic preparedness.

## Methods

**Ethics**. The COG-UK study protocol was approved by the Public Health England Research Ethics Governance Group (reference: R&D NR0195). Public Health England affiliated authors had access to identifiable Cambridgeshire community case data. This data was processed under Regulation 3 of The Health Service (Control of Patient Information) Regulations 2002- permitting the processing of confidential patient information for communicable disease and other risks to public health and as such, individual patient consent is not required. Other authors only had access to anonymised or summarised data. Ethical approval for the UoC asymptomatic COVID-19

screening programme was granted by the UoC Human Biology Research Ethics Committee (HBREC.2020.35) with informed consent gained from participants.

**Study setting**. The UoC has ~23,000 students and 12,600 staff. The university is divided into 31 colleges and 150 departments, faculties and other institutions. Students belong to a college community, as well as being members of the university and an academic faculty/department. Colleges provide residential accommodation for approximately two thirds of students, either on campuses or in off-site housing, and offer social and sports activities, pastoral and academic support for each individual[24]. All colleges have membership from students across multiple courses. The university is based in the City of Cambridge (which has an estimated population of 123,900[25]), in the county of Cambridgeshire (estimated population 855,796 people in 2019[26]) in the East of England.

**Participants and samples**. Samples were derived from university symptomatic testing and asymptomatic COVID-19 screening programmes between 5 October 2020 and 6 December 2020, covering the full term. Testing for all symptomatic students and staff was available on weekdays. The asymptomatic screening programme has been described in detail elsewhere[11]. In brief, screening was offered on a voluntary basis to all students residing in accommodation owned or managed by a college or the Cambridge Theological Federation. In total, 15,561 students were eligible to participate. To optimise testing efficiency, multiple swabs were pooled into the same tube of viral transport medium at the time of sample collection. Testing pools varied in size from 1 to 10 students, with each devised to include one or more student households as far as possible[11]. In this study, households are defined as individuals who share a kitchen, bathroom and/or lounge facilities. The members of any pool testing positive were re-tested using individual confirmatory PCR tests to confirm the result and identify the positive subject(s) (see Supplementary Methods for further details including infection prevention control measures). Only samples from individuals that were confirmed positive upon the re-testing were used for sequencing.

SARS-CoV-2 strains circulating in the local community were identified from the COG-UK dataset for Cambridgeshire. These data were derived from local community samples from non-hospitalised, symptomatic individuals, who requested a free diagnostic test via national community testing. Other samples were derived from patients treated at three Cambridgeshire hospital trusts: Cambridge University Hospitals NHS Foundation Trust (a teaching hospital providing secondary care services for Cambridge and the surrounding area as well as tertiary referral services for the East of England and surge capacity for COVID-19); Royal Papworth Hospital NHS Foundation Trust (specialist heart and lung hospital, also providing surge capacity for COVID-19); Cambridgeshire and Peterborough NHS Foundation Trust (provider of community, mental health and learning disability services in Cambridgeshire). Hospital samples were obtained from both asymptomatic screening and those exhibiting COVID-19 symptoms. Finally, samples were derived from the asymptomatic HCW programme at Cambridge University Hospitals[27].

**Sequencing**. Positive samples from UoC testing with a PCR cycle threshold value ≤33 were selected and sequenced using the GridION platform (Oxford Nanopore). All Cambridgeshire samples sequenced between 24th September and 21st December 2020 were included to overlap with the university term. Samples from the local Cambridgeshire community and hospital cases (described above) were collected as part of national SARS-CoV-2 testing, and sequenced at one of seventeen COG-UK sequencing sites (further details in Supplementary Methods). The samples were prepared using either the ARTIC[28] or veSeq[29] protocols, and were sequenced using Illumina or Oxford Nanopore platforms. Genomic data were filtered to exclude sequences with >5% Ns and those of spuriously low file sizes (<29 KB). Genomes were aligned with minimap2[30] to the Wuhan Hu-1 reference genome (MN908947.3), collected December 2019. All samples were processed through COVID-CLIMB pipelines[31,32]. Protocols are available at https://github.com/COG-UK.

**Phylogenetic analysis**. Maximum likelihood phylogenetic trees were estimated using IQ-TREE (version 2.1.2 COVID-edition)[33] and rooted using Wuhan Hu-1 (MN908947.3) as an outgroup. Trees were constructed using the GTR + Γ substitution model[34], as determined by ModelFinder[35]. Branch support statistics were generated using the ultrafast bootstrap method[36]. TempEst[37] was used to explore the temporal signal in the data. Trees were visualised, explored, and labelled with associated metadata using Microreact[38] to identify epidemiological links supported by the genomic data. Specified mutations were identified using type_variants (https://github.com/cov-ert/type_variants). Possible transmission clusters were defined by extracting phylogenetic neighbourhoods identified using the CIVET tool (version 2.1.0) on 11 January 2021 (https://github.com/artic-network/civet). In selected clusters, further evaluation was conducted using A2B-COVID[15]. A2B-COVID evaluates data from individuals in a pairwise manner. Using viral genome sequences from two individuals, alongside data describing the timing of infection, it evaluates whether or not these data are consistent with a hypothesis that SARS-CoV-2 was transmitted directly from one individual to the other; data from each pair is described as being either consistent, borderline, or unlikely to have been observed given this hypothesis. Where indicated, collapsed nodes from trees generated from CIVET were inspected to visualise data in the context of the COG-UK national database (https://www.cogconsortium.uk/). For further evaluation of

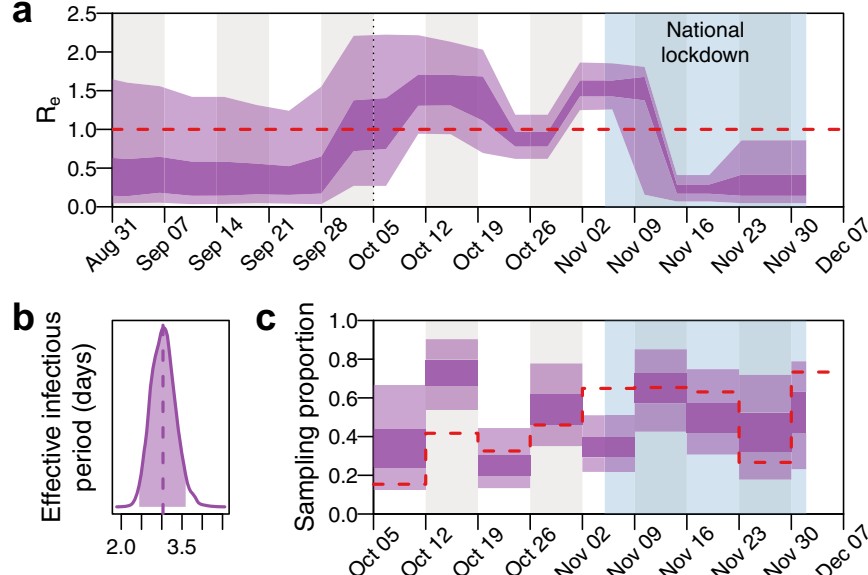

**Fig. 5 Effective reproduction number and infectious period of SARS-CoV-2 from a dominant university cluster.** A 20-epoch birth-death skyline model shows the effect of local infection control measures and the national lockdown on the effective reproduction number ($R_e$), and estimates of the mean effective infectious period as 3.03 (95% HPD = 2.44-3.59) days. **a** $R_e$ posterior estimates (dark shading = 50% HPD; light shading = 95% HPD). The dotted line indicates the start of term and the light blue shaded area the 4-week national lockdown in the UK, which was associated with a large fall in COVID-19 cases in University students. The red dashed line indicates $R_e = 1$. **b** Effective infectious period posterior estimates (shaded region = 95% HPD; dashed line = median). **c** Weekly sampling proportion posterior estimates (dark shading = 50% HPD; light shading = 95% HPD). The red dashed line indicates the empirical sampling proportion estimates for each week in term (number of sequenced genomes from all University clusters divided by the number of positive tests among University staff and students). Source data are provided as a Source Data file.

transmission in the largest cluster identified by CIVET, pairwise SNP differences between sequences were determined using SNP-dist (https://github.com/tseemann/snp-dists/releases/tag/v0.7.0).

**Lineages.** Global Pango Lineages[39] were assigned to each genome using Pangolin (https://github.com/cov-lineages/pangolin/releases/tag/v2.1.6) with analyses performed on COVID-CLIMB[32] (further details in Supplementary Methods).

**Molecular clock and phylodynamic analyses.** BEAST v1.10.4[40] was used to perform a time-scaled phylogenetic analysis using an exponential growth coalescent treeprior and a GTR + Γ substitution model including all university and community high-quality genomes from the study period. As there was a lack of clear temporal signal in our dataset due to the relatively short time period analysed, the substitution rate was fixed to $8 \times 10^{-4}$ substitutions per site per year (s/s/y) under a strict clock model in line with previous SARS-CoV-2 analyses[13,41–44]. Two chains of 100 million iterations were run independently to ensure convergence to the correct posterior distribution. Convergence was assessed using Tracer[45], and 10% of states were removed to account for burn-in. Finally, a maximum clade credibility (MCC) tree was generated using TreeAnnotator.

To estimate the effective reproduction number ($R_e$) and infectious period of SARS-CoV-2 over the term, a dominant clade (representing 69.6% of all university genomes) was selected and all community genome sequences that cluster with it incorporated, resulting in a total of 354 genomes. A Bayesian birth-death skyline (BDSKY) model[16] was employed using BEAST v2.6[46]. A GTR + Γ substitution model was used along with placing a lognormal prior also with mean $8 \times 10^{-4}$ s/s/y (in real space) and standard deviation 0.1 on the clock rate. A lognormal prior with mean 0 and standard deviation 1 was placed on $R_e$ and a Beta prior with $a = 5$ and $\beta = 5$ was placed on the sampling proportion. $R_e$ was parameterised into 20 epochs, equidistantly spaced between the origin time and the most recent sequence collection date. The sampling proportion was fixed to 0 before the first week of term and estimated for each week thereafter. The rate at which infected patients become non-infectious was assumed to be constant and a lognormal prior with mean 48.7 years$^{-1}$ (in real space) and standard deviation 0.25 was placed on it, resulting in a prior mean effective infectious period between ~5 and ~15 days. To test the robustness of the posterior estimates different parameterisations were used for $R_e$ and the sampling proportion, and the sampling proportion prior was varied. Further details are provided in the supplementary methods. To test the robustness of posterior estimates to the clock rate prior all analyses were repeated using a lognormal prior with mean $1 \times 10^{-3}$ s/s/y (in real space) and standard deviation 0.1 on the clock rate. Finally, to test the assumption of a strict clock model, analyses were repeated using an uncorrelated lognormally distributed relaxed clock model[47]. In these analyses the 95% HPD interval of the

coefficient of variation of the clock rate did not exclude 0, indicating poor support for a relaxed clock model in this dataset. Furthermore, estimates of the BDSKY model parameters did not differ significantly from estimates under a strict clock model. Therefore, we only show results under a strict clock model. For all models three chains of 200 million iterations were run independently. Convergence was assessed using the R-package coda[48], and 10% of states were removed to account for burn-in. MCC trees were generated using TreeAnnotator.

**Household attack rates.** A2B-COVID[15] was used to exclude households for which the sequence and epidemiological data were inconsistent with a single viral introduction to the household. A chain binomial model was then used to estimate the probability that an infected person transmitted the virus to an uninfected person within the same household (further details in supplementary methods).

**Epidemiological data.** University student demographic data were derived from the UoC student electronic record system CamSIS, and household structure and membership data from the UoC asymptomatic screening programme. To identify university affiliated cases (students and staff) and hospital staff accessing the national SARS-CoV-2 testing service, Second Generation Surveillance System (SGSS) and contact-tracing data provided by NHS Test and Trace (T&T) data were interrogated. Epidemiologically linked common exposures for students, university staff and the local community were identified through T&T data. Common exposures were defined by T&T as locations or events that two or more people testing positive for COVID-19 visited in the same two to seven day period before symptom onset or positive test. Additional contact tracing information was also provided by the UoC COVID helpdesk. These data were compared with observed phylogenetic clusters to determine potential sources of transmission and determine the extent of transmission between the university and community.

Epidemiological data from UoC were initially compiled in Microsoft Azure SQL and Excel 2013 (Microsoft) and analysed in STATA 14.2 (College Station, TX, USA). Further data manipulation, statistical analysis and figure generation was undertaken with RStudio (version 1.3.1093) using R (version 4.0.2). Network diagrams were produced with R package iGraph (v1.2.6).

**Reporting summary.** Further information on research design is available in the Nature Research Reporting Summary linked to this article.

## Data availability

The Assembled/consensus genomes generated in this study have been deposited in the GISAID[49] database and raw reads are available from European Nucleotide Archive

(ENA)[50] under accession PRJEB37886. Pooled sample sequence raw reads and assembled sequences are deposited in the NCBI Sequence Read Archive Database (SRA; https://www.ncbi.nlm.nih.gov/sra) under the BioProject accession number PRJNA779279.

ENA and Genbank accession codes for individual sequences used in this study are available in supplementary materials (Supplementary Data 1 and 2). All genomes, phylogenetic trees and basic metadata are available from the COG-UK consortium website (https://www.cogconsortium.uk/data). Limited public metadata, analysis files, and processed genomic data for this work are available from GitHub at https://github.com/COG-UK/camb-uni-phylo/ (https://doi.org/10.5281/zenodo.5643354[51]), which also contains a list of ENA and Genbank study sequence accession numbers for this study. For confidentiality reasons, extended metadata[52] are under restricted access for confidentiality reasons and in line with study ethics; requests for access should be directed to corresponding authors and specifically for Public Health England data, to the Public Health England office of data release (https://www.gov.uk/government/publications/accessing-public-health-england-data/about-the-phe-odr-and-accessing-data) with an estimated 60 working days turnaround time. Processed metadata generated for figures in this study are provided in the Source Data file. Source data are provided with this paper.

## Code availability

Custom code used in this analysis is available at https://github.com/COG-UK/camb-uni-phylo/. Please direct further queries to the corresponding authors.

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

## Acknowledgements

Authors A.S.J., W.H. and T.F., and authors L.d.P. and V.H. contributed equally. We thank members of the COVID-19 Genomics Consortium UK and NHS Test and Trace contact tracers for their contributions to generating data used in this study. We thank the Sanger Covid Team for assisting with Samples and Logistics. We are grateful to all students and staff at the University of Cambridge who have contributed to the COVID-19 response during Michaelmas Term. We are grateful to all staff members of the Cambridge COVID-19 Testing Centre for generating qPCR data. D.A. is a Wellcome Clinical PhD Fellow and gratefully supported by the Wellcome Trust (Grant number: 222903/Z/21/Z). B.W. receives funding from the University of Cambridge and the National Institute for Health Research (NIHR) Cambridge Biomedical Research Centre (BRC) at the Cambridge University Hospitals NHS Foundation Trust. I.G. is a Wellcome Senior Fellow and is supported by the Wellcome Trust (Grant number: 207498/Z/17/Z and 206298/B/17/Z). E.M.H. is supported by a UK Research and Innovation (UKRI) Fellowship: MR/S00291X/1. C.J.R.I. acknowledges Medical Research Council (MRC) funding (ref: MC_UU_00002/11). NJM is supported by the MRC (CSF MR/P008801/1) and NHSBT (WPA15-02). A.J.P. gratefully acknowledge the support of the Biotechnology and Biological Sciences Research Council (BBSRC); their research was funded by the BBSRC Institute Strategic Programme Microbes in the Food Chain BB/R012504/1 and its constituent project BBS/E/F/000PR10352, also Quadram Institute Bioscience BBSRC funded Core Capability Grant (project number BB/CCG1860/1). L.d.P. and O.G.P. were supported by the Oxford Martin School. This research was supported by the NIHR Cambridge BRC. The views expressed are those of the authors and not necessarily those of the NHS, the NIHR, or the Department of Health and Social Care. The COVID-19 Genomics UK Consortium is supported by funding from the MRC part of UK Research & Innovation (UKRI), the National Institute of Health Research and Genome Research Limited, operating as the Wellcome Sanger Institute. The Cambridge Covid-19 testing Centre is funded by the Department of Health and Social Care, UK Government. The funders had no role in study design, data collection and analysis, decision to publish, or preparation of the manuscript. For the purpose of Open Access, the author has applied a CC-BY public copyright licence to any Author Accepted Manuscript version arising from this submission.

## Author contributions

All authors read the manuscript and consented to its publication. Where a = conceptualization; b = methodology; c = software; d = validation; e = formal analysis; f = investigation; g = resources; h = data curation; i = writing—original draft preparation; j = writing—review and editing; k = visualization; l = supervision; m = project administration; n = funding acquisition, D.A. contributed a, b, c, d, e, f, h, i, j, k, m; B.B. contributed f, g, h, j, m; CB f, g, h; D.D.A. contributed j, l; G.D. contributed l; L.d.P. contributed b, c, e, g, h, j, k; T.F. contributed a, b, d, e, f, h, j; S.F. contributed f, g, h; I.G. contributed e, f, h; Y.C. contributed e, f, h; I.G.G. contributed a, b, f, g, j, l, m, n; Gr.H. contributed e, f, h; W.L.H. contributed a, b, c, e, j, k; E.M.H. contributed a, b, g, j, l, m, n; V.H. contributed b, c, e, j; J.H. contributed f, g, h; M.H. contributed e, f, h; R.H. contributed b, d, g, h; Ga.H. contributed f, h; R.I. contributed e, f, h; C.I. contributed b, c, e, h, j, k, l; A.J. contributed b, e, f, h, j; D.L contributed f; C.L. contributed a, g, j, n; D.M. contributed g, l, m; N.J.M. contributed a, b, j, l, m, n; P.H.M. contributed j, l, m, n; M.M. contributed f, g, h; R.M. contributed j, l, m; O.N. contributed f, h; S.P. contributed b, d, g, h; A.J.P. contributed l; S.J.P. contributed b, g, j, l, m, n; M.L.P. contributed e, f, h; O.G.P. contributed j, l; B.W. contributed a, b, d, e, f, h, j, k, m; Em.W. contributed a, j; G.W. contributed f, g, h; M.W. contributed j, m, n; El.W. contributed f, h, j; Cambridge Covid-19 Testing Centre contributed b, d, g, h; University of Cambridge Asymptomatic COVID-19 Screening Programme Consortium contributed b, d, g, h; The COVID-19 Genomics UK (COG-UK) Consortium contributed b, d, g, h.

## Competing interests

R.H. is an employee of AstraZeneca AB. The remaining authors declare no competing interests.

## Additional information

[1]University of Cambridge, Department of Medicine, Cambridge, UK. [2]Public Health England, 61 Colindale Ave, London NW9 5EQ, UK. [3]Cambridge University Hospital NHS Foundation Trust, Cambridge, UK. [4]Wellcome Sanger Institute, Hinxton, Cambridge, UK. [5]Cambridge Institute for Therapeutic Immunology and Infectious Disease, University of Cambridge, Cambridge, UK. [6]University of Cambridge, Department of Pathology, Division of Virology, Cambridge, UK. [7]Department of Zoology, University of Oxford, Oxford, UK. [8]Institute of Evolutionary Virology, University of Edinburgh, Edinburgh, UK. [9]Public Health Directorate, Cambridgeshire County Council and Peterborough City Council, Peterborough, UK. [10]Quadram Institute Bioscience, Norwich Research Park, Norwich NR4 7UQ, UK. [11]Health, Safety & Regulated Facilities Division, University of Cambridge, Cambridge, UK. [12]University Information Services, University of Cambridge, Cambridge, UK. [13]Cambridge Covid-19 Testing Centre, Discovery Sciences, R&D, AstraZenenca, Cambridge, UK. [14]Charles River Laboratories, Chesterford Research Park, Saffron Walden CB10 1XL, UK. [15]Institute for Manufacturing, University of Cambridge, Cambridge, UK. [16]MRC Biostatistics Unit, University of Cambridge, East Forvie Building, Forvie Site, Robinson Way, Cambridge CB2 0SR, UK. [17]Cambridge Institute for Medical Research, University of Cambridge, Cambridge, UK. [18]Department of Applied Mathematics and Theoretical Physics, University of Cambridge, Cambridge, UK. [19]MRC-University of Glasgow Centre for Virus Research, Glasgow, UK. [20]Department of Public Health and Primary Care, University of Cambridge, Cambridge, UK. [21]NHS Blood and Transplant, Cambridge, UK. [128]These authors contributed equally: Dinesh Aggarwal, Ben Warne. [129]These authors jointly supervised this work: Michael P. Weekes, Chris Illingworth, Ewan M. Harrison, Nicholas J. Matheson, Ian G. Goodfellow. *Lists of authors and their affiliations appear at the end of the paper. ✉email: dinesh.aggarwal@nhs.net; eh6@sanger.ac.uk; njm25@cam.ac.uk; ig299@cam.ac.uk

## The Cambridge Covid-19 testing Centre

Rob Howes[13], Alexandra Orton[13], Julie Douthwaite[13], Steve Rees[13], Christopher Brown[13,14], Roger Clark[14], Daniel R. Jones[14], Fred Kuenzi[14], Jennifer Rankin[14] & Ian Waddell[14]

## University of Cambridge Asymptomatic COVID-19 Screening Programme Consortium

Patrick Maxwell[1,3], Nicholas Matheson[1,3,5,21], Chris Abell[22], Vickie Braithwaite[9,23], Craig Brierley[24], Jon Crowcroft[22], Aastha Dahal[25], Kathryn Faulkner[9], Michael Glover[22], Ian Goodfellow[6], Jane Greatorex[26], Jon Holgate[12], Rob Howes[13], Laura James[22], Paul Lehner[1,3,5], Ian Leslie[12], Kathleen Liddell[27], Ben Margolis[25], Duncan McFarlane[15], Sally Morgan[28], Linda Sheridan[9], Sally Valletta[22], Anna Vignoles[22], Martin Vinnell[29], Ben Warne[1,3,5,128], Michael P. Weekes[3,17,129], Mark Wills[1], Stewart Fuller[1], Marina Metaxaki[1], Sarah Hilborne[1], Sarah Berry[1], Mahin Bagheri Kahkeshi[1], Dawn Hancock[1], Jennifer Winster[1], Jessica Enright[30], Richard Samworth[31], Vijay Samtani[12], Gabriela Ahmadi-Assalemi[12], Tom Feather[12], Robin Goodall[12], Steve Hoensch[12], Dean Johnson[12], Martin Hunt[12], Nick Mathieson[12], Katya Nikitina[12], Zara Sheldrake[12], Martin Keen[32], Aris Sato[32], David Connor[32], Jonathan Tolhurst[32], Jack Williman[32], Victoria Hollamby[1], Gillian Weale[11], Sinead Jordan[33], Tania Fatseas[33], Peter Taylor[33], Christine Georgiou[33], Michelle Caspersz[33], Claire McNulty[33], Richard Davies[33], Rebecca Clarke[15], Darius Danaei[15], Rory Dyer[15], Rob Glew[15], Oliver Lambson[15], Karen Gibbs[22], Barbara Mozdzen[22], Gabor Raub[22], Asako Radecki[22], Phil White[22], Robert Hughes[22], Lucie Gransden[22], Matt Ceaser[22], Robert Sing[22], Karl Wilson[22], Ajith Parlikad[15], Maharshi Dhada[15], Tom Ridgman[15], Diane Mungovan[34], Steve Matthews[34], Paul Searle[34], John Mills[34], Andy Neely[35], Robert Henderson[28], Edna Murphy[36], Matthew Russell[36], Anthony Freeling[28], Steve Poppitt[37], Jo Tynan[37], James Knapton[38], Filippo Marchetti[38], Sharon J. Peacock[1,3], Ian G. Goodfellow[6,129✉], Ewan M. Harrison[1,2,4,20,129✉], Dinesh Aggarwal[1,2,3,4,128✉], Thomas Fieldman[1,3], Beth Blane[1], Yasmin Chaudhry[6], Daniela De Angelis[16], Theresa Feltwell[2], Iliana Georgana[6], Nazreen F. Hadjirin[1], Grant Hall[6], William L. Hamilton[1,3,4], Myra Hosmillo[3,6], Chris Illingworth[16,18,19,129], Aminu Jahun[6], Danielle Leek[1], Catherine Ludden[1,2], Malte Pinckert[6], Ashley Shaw[3], Afzal Chaudhry[3], Nicholas M. Brown[3], Lenette Mactavous[3], Sophie Hannan[3], Aleksandra Hosaja[3], Clare Leong[3], Jo Wright[39], Natalie Quinnell[39], Chris Workman[39], Mark Ferris[39], Giles Wright[39], Emmeline Watkins[9] & Elizabeth Wright[9]

[22]University of Cambridge, Cambridge, UK. [23]MRC Epidemiology Unit, University of Cambridge, Cambridge, UK. [24]Office of External Affairs and Communications, University of Cambridge, Cambridge, UK. [25]Cambridge Students' Union, Cambridge, UK. [26]Lucy Cavendish College, University of Cambridge, Cambridge, UK. [27]Centre for Law, Medicine and Life Sciences, Faculty of Law, University of Cambridge, Cambridge, UK. [28]Fitzwilliam College, University of Cambridge, Cambridge, UK. [29]Occupational Health and Safety Services, University of Cambridge, Cambridge, UK. [30]School of Computing Science, University of Glasgow, Glasgow, UK. [31]Statistical Laboratory, Centre for Mathematical Sciences, University of Cambridge, Cambridge, UK. [32]Clinical School Computing Service, School of Clinical Medicine, University of Cambridge, Cambridge, UK. [33]COVID-19 Operations Centre, University of Cambridge, Cambridge, UK. [34]University Messenger Service, University of Cambridge, Cambridge, UK. [35]Vice Chancellor's Office, University of Cambridge, Cambridge, UK. [36]Office of Intercollegiate Services Ltd., Cambridge, UK. [37]St John's College, University of Cambridge, Cambridge, UK. [38]Governance and Compliance Division, University of Cambridge, Cambridge, UK. [39]Occupational Health and Wellbeing, Cambridge University Hospitals NHS Foundation Trust, Cambridge, UK.

## The COVID-19 Genomics UK (COG-UK) Consortium

Dinesh Aggarwal[1], Beth Blane[1], Ellena Brooks[1], Alessandro M. Carabelli[1], Carol M. Churcher[1], Katerina Galai[1], Sophia T. Girgis[1], Ravi K. Gupta[1], Nazreen F. Hadjirin[1], Danielle Leek[1], Catherine Ludden[1,2], Georgina M. McManus[1], Sophie Palmer[1], Sharon J. Peacock[1,2,3,4], Kim S. Smith[1], Elias Allara[2], David Bibby[2], Chloe Bishop[2], Andrew Bosworth[2], Daniel Bradshaw[2], Vicki Chalker[2], Meera Chand[2], Gavin Dabrera[2], Nicholas Ellaby[2], Eileen Gallagher[2], Natalie Groves[2], Ian Harrison[2], Hassan Hartman[2], Richard Hopes[2], Jonathan Hubb[2], Stephanie Hutchings[2], Angie Lackenby[2], Juan Ledesma[2], David Lee[2], Nikos Manesis[2], Carmen Manso[2], Tamyo Mbisa[2], Shahjahan Miah[2], Peter Muir[2], Richard Myers[2], Husam Osman[2], Vineet Patel[2],

Clare Pearson[2], Steven Platt[2], Hannah M. Pymont[2], Mary Ramsay[2], Esther Robinson[2], Ulf Schaefer[2], Alicia Thornton[2], Katherine A. Twohig[2], Ian B. Vipond[2], David Williams[2], William L. Hamilton[1,3,4], Ben Warne[1,3,5,128], Louise Aigrain[4], Alex Alderton[4], Roberto Amato[4], Cristina V. Ariani[4], Jeff Barrett[4], Andrew R. Bassett[4], Mathew A. Beale[4], Charlotte Beaver[4], Katherine L. Bellis[4], Emma Betteridge[4], James Bonfield[4], Iraad F. Bronner[4], Michael H. S. Chapman[4], John Danesh[4], Robert Davies[4], Matthew J. Dorman[4], Eleanor Drury[4], Jillian Durham[4], Ben W. Farr[4], Luke Foulser[4], Sonia Goncalves[4], Scott Goodwin[4], Marina Gourtovaia[4], Ewan M. Harrison[1,2,4,20,129✉], David K. Jackson[4], Keith James[4], Dorota Jamrozy[4], Ian Johnston[4], Leanne Kane[4], Sally Kay[4], Jon-Paul Keatley[4], Dominic Kwiatkowski[4], Cordelia F. Langford[4], Mara Lawniczak[4], Stefanie V. Lensing[4], Steven Leonard[4], Laura Letchford[4], Kevin Lewis[4], Jennifer Liddle[4], Rich Livett[4], Stephanie Lo[4], Alex Makunin[4], Inigo Martincorena[4], Shane McCarthy[4], Samantha McGuigan[4], Robin J. Moll[4], Rachel Nelson[4], Karen Oliver[4], Steve Palmer[4], Naomi R. Park[4], Minal Patel[4], Liam Prestwood[4], Christoph Puethe[4], Michael A. Quail[4], Diana Rajan[4], Shavanthi Rajatileka[4], Nicholas M. Redshaw[4], Carol Scott[4], Lesley Shirley[4], John Sillitoe[4], Scott A. J. Thurston[4], Gerry Tonkin-Hill[4], Jaime M. Tovar-Corona[4], Danni Weldon[4], Andrew Whitwham[4], Yasmin Chaudhry[6], Iliana Georgana[6], Ian G. Goodfellow[6,129✉], Grant Hall[6], Myra Hosmillo[3,6], Aminu S. Jahun[6], Malte L. Pinckert[6], Stephen W. Attwood[7], Louis du Plessis[7], Marina Escalera Zamudio[7], Sarah Francois[7], Bernardo Gutierrez[7], Moritz U. G. Kraemer[7], Oliver G. Pybus[7], Jayna Raghwani[7], Tetyana I. Vasylyeva[7], Alex E. Zarebski[7], Verity Hill[8], Nabil-Fareed Alikhan[10], Alp Aydin[10], David J. Baker[10], Leonardo de Oliveira Martins[10], Gemma L. Kay[10], Thanh Le-Viet[10], Alison E. Mather[10], Lizzie Meadows[10], Justin O'Grady[10], Andrew J. Page[10], Steven Rudder[10], Alexander J. Trotter[10], Chris J. Illingworth[16], Chris Jackson[16], Elihu Aranday-Cortes[19], Patawee Asamaphan[19], Alice Broos[19], Stephen N. Carmichael[19], Ana da Silva Filipe[19], Joseph Hughes[19], Natasha G. Jesudason[19], Natasha Johnson[19], Kathy K. Li[19], Daniel Mair[19], Jenna Nichols[19], Seema Nickbakhsh[19], Marc O. Niebel[19], Kyriaki Nomikou[19], Richard J. Orton[19], David L. Robertson[19], Rajiv N. Shah[19], James G. Shepherd[19], Joshua B. Singer[19], Igor Starinskij[19], Emma C. Thomson[19], Lily Tong[19], Sreenu Vattipally[19], Amy Ash[40], Cherian Koshy[40], Nick Cortes[41], Stephen Kidd[41], Jessica Lynch[41], Nathan Moore[41], Matilde Mori[41], Emma Wise[41], Tanya Curran[42], Derek J. Fairley[42], James P. McKenna[42], Helen Adams[43], David Bonsall[44], Christophe Fraser[44], Tanya Golubchik[44], Benjamin J. Cogger[45], Mohammed O. Hassan-Ibrahim[45], Cassandra S. Malone[45], Nicola Reynolds[46], Michelle Wantoch[46], Safiah Afifi[47], Robert Beer[47], Michaela John[47], Joshua Maksimovic[47], Kathryn McCluggage[47], Sian Morgan[47], Karla Spellman[47], Catherine Bresner[48], Thomas R. Connor[48], William Fuller[48], Martyn Guest[48], Huw Gulliver[48], Christine Kitchen[48], Angela Marchbank[48], Ian Merrick[48], Robert Munn[48], Anna Price[48], Joel Southgate[48], Trudy Workman[48], Amita Patel[49], Luke B. Snell[49], Rahul Batra[50], Themoula Charalampous[50], Jonathan Edgeworth[50], Gaia Nebbia[50], Angela H. Beckett[51], Samuel C. Robson[51], David M. Aanensen[52], Khalil Abudahab[52], Mirko Menegazzo[52], Ben E. W. Taylor[52], Anthony P. Underwood[52], Corin A. Yeats[52], Louise Berry[53], Tim Boswell[53], Gemma Clark[53], Vicki M. Fleming[53], Hannah C. Howson-Wells[53], Carl Jones[53], Amelia Joseph[53], Manjinder Khakh[53], Michelle M. Lister[53], Wendy Smith[53], Iona Willingham[53], Paul Bird[54], Karlie Fallon[54], Thomas Helmer[54], Christopher Holmes[54], Julian Tang[54], Victoria Blakey[55], Sharon Campbell[55], Veena Raviprakash[55], Nicola Sheriff[55], Lesley-Anne Williams[55], Matthew Carlile[56], Johnny Debebe[56], Nadine Holmes[56], Matthew W. Loose[56], Christopher Moore[56], Fei Sang[56], Victoria Wright[56], Francesc Coll[57], Gilberto Betancor[58], Adrian W. Signell[58], Harry D. Wilson[58], Thomas Davis[59], Sahar Eldirdiri[59], Anita Kenyon[59], M. Estee Torok[60], Hannah Lowe[61], Samuel Moses[61], Luke Bedford[62], Jonathan Moore[63], Susanne Stonehouse[63], Ali R. Awan[64], Chloe L. Fisher[64],

John BoYes[65], Laura Atkinson[66], Judith Breuer[66], Julianne R. Brown[66], Kathryn A. Harris[66], Jack C. D. Lee[66], Divya Shah[66], Nathaniel Storey[66], Flavia Flaviani[67], Adela Alcolea-Medina[68], Gabrielle Vernet[69], Rebecca Williams[69], Michael R. Chapman[70], Wendy Chatterton[71], Judith Heaney[71], Lisa J. Levett[71], Monika Pusok[71], Li Xu-McCrae[72], Matthew Bashton[73], Darren L. Smith[73], Gregory R. Young[73], Frances Bolt[74], Alison Cox[74], Alison Holmes[74], Pinglawathee Madona[74], Siddharth Mookerjee[74], James Price[74], Paul A. Randell[74], Olivia Boyd[75], Fabricia F. Nascimento[75], Lily Geidelberg[75], Rob Johnson[75], David Jorgensen[75], Manon Ragonnet-Cronin[75], Aileen Rowan[75], Igor Siveroni[75], Graham P. Taylor[75], Erik M. Volz[75], Katherine L. Smollett[76], Nicholas J. Loman[77], Claire McMurray[77], Alan McNally[77], Sam Nicholls[77], Radoslaw Poplawski[77], Joshua Quick[77], Will Rowe[77], Joanne Stockton[77], Rocio T. Martinez Nunez[78], Cassie Breen[79], Angela Cowell[79], Jenifer Mason[79], Elaine O'Toole[79], Trevor I. Robinson[79], Joanne Watts[79], Graciela Sluga[80], Shazaad S. Y. Ahmad[81], Ryan P. George[81], Nicholas W. Machin[81], Fenella Halstead[82], Wendy Hogsden[82], Venkat Sivaprakasam[82], Holli Carden[83], Antony D. Hale[83], Katherine L. Harper[83], Louissa R. Macfarlane-Smith[83], Shirelle Burton-Fanning[84], Jennifer Collins[84], Gary Eltringham[84], Brendan AI. Payne[84], Yusri Taha[84], Sheila Waugh[84], Sarah O'Brien[85], Steven Rushton[85], Rachel Blacow[86], Amanda Bradley[86], Alasdair Maclean[86], Guy Mollett[86], Rebecca Dewar[87], Martin P. McHugh[87], Kate E. Templeton[87], Elizabeth Wastenge[87], Lindsay Coupland[88], Samir Dervisevic[88], Emma J. Meader[88], Rachael Stanley[88], Louise Smith[89], Edward Barton[90], Clive Graham[90], Debra Padgett[90], Garren Scott[90], Jane Greenaway[91], Emma Swindells[91], Clare M. McCann[92], Andrew Nelson[92], Wen C. Yew[92], Monique Andersson[93], Derrick Crook[93], David Eyre[93], Anita Justice[93], Timothy Peto[93], Nichola Duckworth[94], Tim J. Sloan[94], Sarah Walsh[94], Kelly Bicknell[95], Anoop J. Chauhan[95], Scott Elliott[95], Sharon Glaysher[95], Robert Impey[95], Allyson Lloyd[95], Sarah Wyllie[95], Nick Levene[96], Lynn Monaghan[96], Declan T. Bradley[97], Tim Wyatt[97], Martin D. Curran[98], Surendra Parmar[98], Matthew T. G. Holden[99], Sharif Shaaban[99], Alexander Adams[100], Hibo Asad[100], Alec Birchley[100], Matthew Bull[100], Jason Coombes[100], Sally Corden[100], Simon Cottrell[100], Noel Craine[100], Michelle Cronin[100], Alisha Davies[100], Elen De Lacy[100], Fatima Downing[100], Sue Edwards[100], Johnathan M. Evans[100], Laia Fina[100], Amy Gaskin[100], Bree Gatica-Wilcox[100], Laura Gifford[100], Lauren Gilbert[100], Lee Graham[100], David Heyburn[100], Ember Hilvers[100], Robin Howe[100], Hannah Jones[100], Rachel Jones[100], Sophie Jones[100], Sara Kumziene-SummerhaYes[100], Caoimhe McKerr[100], Catherine Moore[100], Mari Morgan[100], Nicole Pacchiarini[100], Malorie Perry[100], Amy Plimmer[100], Sara Rey[100], Giri Shankar[100], Sarah Taylor[100], Joanne Watkins[100], Chris Williams[100], Anna Casey[101], Liz Ratcliffe[101], Erwan Acheson[102], Zoltan Molnar[102], David A. Simpson[102], Thomas Thompson[102], Cressida Auckland[103], Sian Ellard[103], Christopher R. Jones[103], Bridget A. Knight[103], Jane A. H. Masoli[103], Tanzina Haque[104], Jennifer Hart[104], Dianne Irish-Tavares[104], Tabitha W. Mahungu[104], Eric Witele[104], Ashok Dadrah[105], Melisa L. Fenton[105], Tranprit Saluja[105], Amanda Symmonds[105], Yann Bourgeois[106], Garry P. Scarlett[106], Kate Cook[107], Hannah Dent[107], Christopher Fearn[107], Salman Goudarzi[107], Katie F. Loveson[107], Hannah Paul[107], Cariad Evans[108], Kate Johnson[108], David G. Partridge[108], Mohammad Raza[108], Paul Baker[109], Stephen Bonner[109], Sarah Essex[109], Steven Liggett[109], Ronan A. Lyons[110], Adhyana I. K. Mahanama[111], Kordo Saeed[111], Buddhini Samaraweera[111], Siona Silveira[111], Eleri Wilson-Davies[111], P. Emanuela[111], Nadua Bayzid[112], Marius Cotic[112], Leah Ensell[112], John A. Hartley[112], Riaz Jannoo[112], Angeliki Karamani[112], Mark Kristiansen[112], Helen L. Lowe[112], Sunando Roy[112], Adam P. Westhorpe[112], Rachel J. Williams[112], Charlotte A. Williams[112], Sarah Jeremiah[113], Jacqui A. Prieto[113], Lisa Berry[114], Dimitris Grammatopoulos[114], Katie Jones[114], Sarojini Pandey[114], Andrew Beggs[115], Alex Richter[115], Fiona Ashcroft[116], Angus Best[116], Liam Crawford[116], Nicola Cumley[116], Megan Mayhew[116], Oliver Megram[116], Jeremy Mirza[116], Emma Moles-Garcia[116],

Benita Percival[116], Giselda Bucca[117], Andrew R. Hesketh[117], Colin P. Smith[117], Rose K. Davidson[118], Carlos E. Balcazar[119], Michael D. Gallagher[119], Áine O'Toole[119], Andrew Rambaut[119], Stefan Rooke[119], Thomas D. Stanton[119], Thomas Williams[119], Kathleen A. Williamson[119], Claire M. Bewshea[120], Audrey Farbos[120], James W. Harrison[120], Aaron R. Jeffries[120], Robin Manley[120], Stephen L. Michell[120], Michelle L. Michelsen[120], Christine M. Sambles[120], David J. Studholme[120], Ben Temperton[120], Joanna Warwick-Dugdale[120], Alistair C. Darby[121], Richard Eccles[121], Matthew Gemmell[121], Richard Gregory[121], Sam T. Haldenby[121], Julian A. Hiscox[121], Margaret Hughes[121], Miren Iturriza-Gomara[121], Kathryn A. Jackson[121], Anita O. Lucaci[121], Charlotte Nelson[121], Steve Paterson[121], Lucille Rainbow[121], Lance Turtle[121], Edith E. Vamos[121], Hermione J. Webster[121], Mark Whitehead[121], Claudia Wierzbicki[121], Adrienn Angyal[122], Rebecca Brown[122], Thushan I. de Silva[122], Timothy M. Freeman[122], Marta Gallis[122], Luke R. Green[122], Danielle C. Groves[122], Alexander J. Keeley[122], Benjamin B. Lindsey[122], Stavroula F. Louka[122], Matthew D. Parker[122], Paul J. Parsons[122], Nikki Smith[122], Rachel M. Tucker[122], Dennis Wang[122], Max Whiteley[122], Matthew Wyles[122], Peijun Zhang[122], Mohammad T. Alam[123], Laura Baxter[123], Hannah E. Bridgewater[123], Paul E. Brown[123], Jeffrey K. J. Cheng[123], Chrystala Constantinidou[123], Lucy R. Frost[123], Sascha Ott[123], Richard Stark[123], Grace Taylor-Joyce[123], Meera Unnikrishnan[123], Alberto C. Cerda[124], Tammy V. Merrill[124], Rebekah E. Wilson[124], Jonathan Ball[125], Joseph G. Chappell[125], Patrick C. McClure[125], Theocharis Tsoleridis[125], David Buck[126], Mariateresa de Cesare[126], Angie Green[126], George MacIntyre-Cockett[126], John A. Todd[126], Amy Trebes[126], Rory N. Gunson[127], Claire Cormie[22], Joana Dias[22], Sally Forrest[22], Harmeet K. Gill[22], Ellen E. Higginson[22], Leanne M. Kermack[22], Mailis Maes[22], Chris Ruis[22], Sushmita Sridhar[22] & Jamie Young[22]

[40]Barking, Havering and Redbridge University Hospitals NHS Trust, Romford, UK. [41]Basingstoke Hospital, Basingstoke, UK. [42]Belfast Health & Social Care Trust, Belfast, UK. [43]Betsi Cadwaladr University Health Board, Bangor, UK. [44]Big Data Institute, Nuffield Department of Medicine, University of Oxford, Oxford, UK. [45]Brighton and Sussex University Hospitals NHS Trust, Brighton, UK. [46]Cambridge Stem Cell Institute, University of Cambridge, Cambridge, UK. [47]Cardiff and Vale University Health Board, Cardiff, UK. [48]Cardiff University, Cardiff, UK. [49]Centre for Clinical Infection & Diagnostics Research, St. Thomas' Hospital and Kings College London, London, UK. [50]Centre for Clinical Infection and Diagnostics Research, Department of Infectious Diseases, Guy's and St Thomas' NHS Foundation Trust, London, UK. [51]Centre for Enzyme Innovation, University of Portsmouth (PORT), Portsmouth, UK. [52]Centre for Genomic Pathogen Surveillance, University of Oxford, Oxford, UK. [53]Clinical Microbiology Department, Queens Medical Centre, Nottingham, UK. [54]Clinical Microbiology, University Hospitals of Leicester NHS Trust, Leicester, UK. [55]County Durham and Darlington NHS Foundation Trust, Darlington, UK. [56]Deep Seq, School of Life Sciences, Queens Medical Centre, University of Nottingham, Nottingham, UK. [57]Department of Infection Biology, Faculty of Infectious & Tropical Diseases, London School of Hygiene & Tropical Medicine, London, UK. [58]Department of Infectious Diseases, King's College London, London, UK. [59]Department of Microbiology, Kettering General Hospital, Kettering, UK. [60]Departments of Infectious Diseases and Microbiology, Cambridge University Hospitals NHS Foundation Trust, Cambridge, UK. [61]East Kent Hospitals University NHS Foundation Trust, Canterbury, UK. [62]East Suffolk and North Essex NHS Foundation Trust, Colchester, UK. [63]Gateshead Health NHS Foundation Trust, Gateshead, UK. [64]Genomics Innovation Unit, Guy's and St. Thomas' NHS Foundation Trust, London, UK. [65]Gloucestershire Hospitals NHS Foundation Trust, Cheltenham, UK. [66]Great Ormond Street Hospital for Children NHS Foundation Trust, London, UK. [67]Guy's and St. Thomas' BRC, London, UK. [68]Guy's and St. Thomas' Hospitals, London, UK. [69]Hampshire Hospitals NHS Foundation Trust, Basingstoke, UK. [70]Health Data Research UK Cambridge, Cambridge, UK. [71]Health Services Laboratories, London, UK. [72]Heartlands Hospital, Birmingham, UK. [73]Hub for Biotechnology in the Built Environment, Northumbria University, Newcastle, UK. [74]Imperial College Hospitals NHS Trust, London, UK. [75]Imperial College London, London, UK. [76]Institute of Biodiversity, Animal Health & Comparative Medicine, Glasgow, UK. [77]Institute of Microbiology and Infection, University of Birmingham, Birmingham, UK. [78]King's College London, London, UK. [79]Liverpool Clinical Laboratories, Liverpool, UK. [80]Maidstone and Tunbridge Wells NHS Trust, Tunbridge Wells, UK. [81]Manchester University NHS Foundation Trust, Manchester, UK. [82]Microbiology Department, Wye Valley NHS Trust, Hereford, UK. [83]National Infection Service, PHE and Leeds Teaching Hospitals Trust, Leeds, UK. [84]Newcastle Hospitals NHS Foundation Trust, Newcastle, UK. [85]Newcastle University, Newcastle, UK. [86]NHS Greater Glasgow and Clyde, Glasgow, UK. [87]NHS Lothian, Edinburgh, UK. [88]Norfolk and Norwich University Hospital, Norwich, UK. [89]Norfolk County Council, Norwich, UK. [90]North Cumbria Integrated Care NHS Foundation Trust, Carlisle, UK. [91]North Tees and Hartlepool NHS Foundation Trust, Stockton on Tees, UK. [92]Northumbria University, Newcastle, UK. [93]Oxford University Hospitals NHS Foundation Trust, Oxford, UK. [94]PathLinks, Northern Lincolnshire & Goole NHS Foundation Trust, Grimsby, UK. [95]Portsmouth Hospitals University NHS Trust, Portsmouth, UK. [96]Princess Alexandra Hospital Microbiology Dept, Harlow, UK. [97]Public Health Agency, Belfast, UK. [98]Public Health England, Clinical Microbiology and Public Health Laboratory, Cambridge, UK. [99]Public Health Scotland, Edinburgh, UK. [100]Public Health Wales NHS Trust, Cardiff, UK. [101]Queen Elizabeth Hospital, London, UK. [102]Queen's University Belfast, Belfast, UK. [103]Royal Devon and Exeter NHS Foundation Trust, Exeter, UK. [104]Royal Free NHS Trust, London, UK. [105]Sandwell and West Birmingham NHS Trust, Birmingham, UK. [106]School of Biological Sciences, University of Portsmouth (PORT), Portsmouth, UK. [107]School of Pharmacy and Biomedical Sciences, University of Portsmouth (PORT), Portsmouth, UK. [108]Sheffield Teaching Hospitals, Sheffield, UK. [109]South Tees Hospitals NHS Foundation Trust, Middlesbrough, UK. [110]Swansea University, Swansea, UK. [111]University Hospitals Southampton NHS Foundation Trust, Southampton, UK. [112]University College London, London, UK. [113]University Hospital Southampton NHS Foundation Trust, Southampton, UK. [114]University Hospitals Coventry and Warwickshire, Coventry, UK. [115]University of Birmingham, Birmingham, UK. [116]University of Birmingham Turnkey Laboratory, Birmingham, UK. [117]University of Brighton, Brighton, UK. [118]University of East Anglia, Norwich, UK. [119]University of Edinburgh, Edinburgh, UK. [120]University of Exeter, Exeter, UK. [121]University of

Liverpool, Liverpool, UK. [122]University of Sheffield, Sheffield, UK. [123]University of Warwick, Warwick, UK. [124]Viapath, Guy's and St Thomas' NHS Foundation Trust, and King's College Hospital NHS Foundation Trust, London, UK. [125]Virology, School of Life Sciences, Queens Medical Centre, University of Nottingham, Nottingham, UK. [126]Wellcome Centre for Human Genetics, Nuffield Department of Medicine, University of Oxford, Oxford, UK. [127]West of Scotland Specialist Virology Centre, NHS Greater Glasgow and Clyde, Glasgow, UK.

