## [Peer Review File · Nature Communications]

Genomic epidemiology of SARS-CoV-2 in a UK university identifies dynamics of transmissionREVIEWER COMMENTS

Reviewer #1 (Remarks to the Author):

In this article, Aggarwal and colleagues investigate the spread of SARS-CoV-2 in the university of Cambridge between Oct and Dec 2020. For this, they take advantage of (i) a dense sampling of viral genomes both from students and staff and from the local community, (ii) very good epidemiological data. Combining both sources of information they show that transmission at university was seeded by very few introductions, one being responsible for a vast majority of subsequent cases via initial transmission linked to multiple social events. Attack rate within student households, although not negligible, was lower than in other contexts, which probably reflects a good compliance to control measures implemented within the university, as also attested by measurable reductions of R_e following local and national measures. Importantly, the outbreak within the university did not spread out much into the local community.

One strength of this article is that it is based on a very interesting data set. High quality paired genomic surveillance and contact tracing/epi data was and is still rare (at least as far as I know). In addition, this dataset focuses on a large, pre-variant outbreak (only a couple of detections of Alpha). As such it provides an interesting point of comparison for future studies (and it must be highlighted that similar points of comparison are unlikely to be generated in the near future: given the current needs, retrospective genomic surveillance will likely remain a luxury in many places).

The analyses produced by the authors seem robust enough and deliver interesting insights into the dynamics of this specific outbreak. Of course, it is hard to say how much can be extrapolated to other higher education settings – this institution presenting unique characteristics (essentially those of an elite school, including e.g. the privileged socioeconomic background of most students) but this limitation is acknowledged by the authors.

Finally, the article is well written and very clear. All in all, I am supportive of its publication in Nature Communications.

I only have one minor comment: the authors present results obtained with AB2-COVID to support that within cluster transmission likely represented continuous transmission (fig 3c). I am not familiar with this method and other readers might be in the same case. It'd be nice to add a brief description in the method section and possibly a couple of sentences to support the interpretation in the fig legend (eg I assume individuals are ordered chronologically?).

Sebastien Calvignac-Spencer

Reviewer #2 (Remarks to the Author):

In "Genomic epidemiology of SARS-CoV-2 in a UK university identifies dynamics of transmission" the authors provide a descriptive study of how SARS-CoV-2 transmitted within a University setting. This setting of high sampling coupled with sequencing, contact tracing and epidemiological investigations provides a mostly controlled environment to study transmission dynamics. The piece is well written and provides valuable insights into the risk factor and mitigation strategies relevant for this younger population who, across the world' will likely be in the latest tranches of vaccination and therefore most reliant on non-pharmaceutical interventions for longer.

I would recommend this for publication after minor revisions as I do feel there are several areas in which the paper could be improved to maximise the impact and clarify some of the findings for a wider audience.

Recommendations for revision:

- For the wide readership of this journal it might be beneficial to include a transmission network visualisation of especially cluster 1 to visualise the repeated 'mass spreading' at the social event and spread (or lack thereof) within households. I.e. all current figures are focused mostly on

cases, in such a high sampling setting it would be helpful to visualise how many individuals did not test at all vs. did not test positive/were not infected.

- The statistical test performed on the 'SNP differences between cases' does not account for the fact that the University samples were geographically and socially more closely related, comparing these to 'randomly' sampled community cases will distort the importance of this observed lack of phylogenetic diversity. If a sufficient sample size is available from a smaller region/sub-region with greater social cohesion that is comparable to the University this would be preferable. If the statement of significance is intended to show genomic similarity as a reflection of 'interaction'/cohesion the statement could be caveat-ed as such to ensure the reader is aware of the underlying population structure differences.
- Line 167-169 and 342-344, while the COG-UK consortium has done a truly momentous amount of work the fact that community genome coverage is usually between 8-12% means that statements like "suggesting that the university cases were introduced from outside Cambridgeshire" would be best caveat-ed with a (Bayesian) probability and confidence interval to illustrate the uncertainty in the data.
- Line 256-259, this statement reads like there were no shared courses/areas/etc.. identified in 7 out of the 8 genomics clusters. What does this mean for the perceived completeness of the data set on both the genomics and epidemiological front? Some wording to explore these cases of missing data vs missing links might be useful. Line 355-358 does not seem to reflect on this finding in the current version.
- There seems to be limited information provided on what statistical/network/other analysis were performed on the epidemiological data and how these could be replicated on other datasets. Would it be possible to share the de-identified contact matrices and household/course data with associated code to enable others to re-create Figure 3c and other epidemiologically informed analysis?
 - o Qualifying what data is considered 'basic' (line 575) would be helpful for people to know where they should go to find/obtain certain data
- It is interesting to see that B.1.1.7, which is considered to spread more easily/rapidly, did not lead to increasing case numbers within the college while it did in the community at the same time. Was there anything particular about the B.1.1.7 cases in the University setting that prevent spread?

Recommendations for minor changes:

- Figure 2 a/c, it might be more intuitive to have all B.1.177.Xs be a different shading of the base lineage colour to reflect the phylodynamic relations between lineages and to retain black or grey for 'other'
- Figure 4 legends of panel B and C appear to be swapped
- Line 148-154, it would be good to include the estimated/inferred number of introductions into the university to contextualise this statement "signifying that the majority of introductions into UoC did not cause ongoing transmission." Currently it is unclear if this is '5 out of 198' or a different number.
- Line 136, it would be useful to know what date/version of Pango this was in text as version changes can have a major impact on the number of lineages possibly identified
- Line 318: "demonstrating both compliance and an effective control strategy." Is it not one leading to the other?
- Methods; when referring to a specific version it is useful to refer to the specific tag like this: <https://github.com/tseemann/snp-dists/releases/tag/v0.7.0>.
 - o In addition, some of the links in the methods section do not seem to be properly hyperlinked.
- In the legends of supplement figures, specifically the CIVET + microreact trees, there is no specification of what the square symbol represents.

Joep de Ligt

Reviewer #3 (Remarks to the Author):

This manuscript presents a detailed, well written, polished outbreak investigation focused on The U of Cambridge between 5 October and 6 December 2020. However, while the paper is well written, the conclusions don't directly flow from the information as it has been presented. While the focus is on the university setting, there is only sporadic, broad information about the specific university COVID-19 control measures in the discussion and supplementary information, and the authors have not really drawn together specific information on the timing and detail of control measures at the university level, and the changes in case numbers/ R_e , in order to make any suggestion of impact. This would have made the paper much more compelling. I suspect that there was not a great deal of difference in the local university control measures over time, and the real change was the implementation of the national lockdown, which is perhaps why this has been glossed over.

Specific comments:

While the manuscript provides a comprehensive, polished genomic epidemiological examination of an outbreak in a university setting which is certainly of local interest, it is difficult to discern the broader implications of the findings presented. Transmission in household, workplace and social settings, including nightclubs, has been well described previously. Without further clarification of what is really unique about this report it is hard to justify publication in this journal.

Abstract, line 75-76. The author's note that the study highlights effective interventions in a higher education setting, but these have not been well described or explored in the manuscript. Suggest authors considering including the extent and, particularly, timing of UoC and national interventions in the main manuscript and how these relate to the observed changes in case numbers and R_e . Without this, the paper is not highlighting any direct interventions that could be used to more safely open up universities in future.

Discussion, lines 314-315. The author's note a rise in reproductive rate following the announcement of national lockdown, and suggest the need for a targeted public health campaign to reduce high risk activities during the 'lag' period. Can the authors justify why they have not suggested that the time between announcement and implementation of the lockdown be reduced? In some settings such actions are implemented ~24 hours following announcement, ensuring interventions are implemented sooner and reducing opportunity for geographical dispersion and increase engagement in high-risk behaviour during this period.

Figure 2c: Can the authors suggest why a steeper decline in SARS-CoV-2 cases was observed within UoC than in the local community, and how the interventions in place at the time of the observed decrease differed between settings?

Conclusion, lines 366-367: The authors conclude that 'the insights gained will inform public policy regarding infection control measures in higher education settings', however limited attention has been paid to the timing and impact of local (and national) control measures. If the author's wish to suggest this as a major conclusion of the paper, I suggest a closer examination of the detail and timing of specific control measures and the changes in R_e /case numbers is provided.

Discussion line 351: The authors advocate for a combined genomic epidemiological approach to outbreak investigations (and presumably surveillance more broadly). Can the authors please comment on if genomic data was used prospectively to inform control measures throughout the study period, or if all genomic analyses were performed retrospectively?

Additional work is required for the modelling aspects of the paper:

First, the authors used ModelFinder to identify the GTR+Gamma model as the most optimal model for the data, yet they chose to use an HKY+Gamma model in their BEAST analyses. The HKY model does not allow for distinct Transition and Transversion rates, which we know are a feature of SARS-CoV-2 data. They need to redo their analyses with a better model. If HKY is more adequate for the BEAST modelling, then this should be outlined clearly for the reader.

Second, the choice of fixing the substitution rate to some generic value from Rambaut et al. 2000

seems unwarranted. There are good estimates for the substitution rate from other papers they could use, for instance Hoshino et al. 2021 *Gene*, Pipes et al. 2021 *MBE* and Duchene et al. 2020 *Viral Evolution*, Seemann et al. 2020 *Nat Comms*. In addition, I would suggest including a bigger period of sequence data from the UK and the rest of the world to get a more robust temporal signal, thus precluding the need to set such a heavy prior on the substitution rate — one that will affect the time estimates of internal nodes and all other derived estimates.

Finally, I think using a strict clock is probably unwarranted, and a relaxed clock is likely to fit the data better. It is good practice to examine different priors on the clock by running models with distinct priors and comparing the models. The best models according to the data can then be used to obtain final estimates of interest (I suggest Stadler et al. 2013 *PNAS* as a good example to follow). The same applies for the BDSK models.

Reviewer 1

In this article, Aggarwal and colleagues investigate the spread of SARS-CoV-2 in the university of Cambridge between Oct and Dec 2020. For this, they take advantage of (i) a dense sampling of viral genomes both from students and staff and from the local community, (ii) very good epidemiological data. Combining both sources of information they show that transmission at university was seeded by very few introductions, one being responsible for a vast majority of subsequent cases via initial transmission linked to multiple social events. Attack rate within student households, although not negligible, was lower than in other contexts, which probably reflects a good compliance to control measures implemented within the university, as also attested by measurable reductions of R_e following local and national measures. Importantly, the outbreak within the university did not spread out much into the local community.

One strength of this article is that it is based on a very interesting data set. High quality paired genomic surveillance and contact tracing/epi data was and is still rare (at least as far as I know). In addition, this dataset focuses on a large, pre-variant outbreak (only a couple of detections of Alpha). As such it provides an interesting point of comparison for future studies (and it must be highlighted that similar points of comparison are unlikely to be generated in the near future: given the current needs, retrospective genomic surveillance will likely remain a luxury in many places).

The analyses produced by the authors seem robust enough and deliver interesting insights into the dynamics of this specific outbreak. Of course, it is hard to say how much can be extrapolated to other higher education settings – this institution presenting unique characteristics (essentially those of an elite school, including e.g. the privileged socioeconomic background of most students) but this limitation is acknowledged by the authors.

Finally, the article is well written and very clear. All in all, I am supportive of its publication in NatureCommunications.

We thank Reviewer 1 for this succinct summary of the critically important findings from this paper and their complimentary assessment of our work, highlighting the value of the study as a detailed analysis of a pre-variant outbreak. We wholeheartedly agree that such a dataset, pre-variants, is unlikely to be made available (and has not been made available to date) by others working in this space. We believe that as sequencing becomes increasingly recognised as pivotal to the control of pandemic, this will be an important piece of research to compare future work to. We thank reviewer 1 for highlighting the robustness of the analyses undertaken in this study. We agree that comparisons to other higher education settings, with potentially different demographics, should be made with caution, as we also highlight in the paper.

I only have one minor comment: the authors present results obtained with AB2-COVID to support that within cluster transmission likely represented continuous transmission (fig 3c). I am not familiar with this method and other readers might be in the same case. It'd be nice to add a brief description in the method section and possibly a couple of sentences to support the interpretation in the fig legend (eg I assume individuals are ordered chronologically?).

We thank Reviewer 1 for pointing this out. We have added a brief description of A2B-COVID in the methods section (lines 484-489):

'A2B-COVID evaluates data from individuals in a pairwise manner. Using viral genome sequences from two individuals, alongside data describing the timing of infection, it evaluates whether or not these data are consistent with a hypothesis that SARS-CoV-2 was transmitted directly from one individual to the other; data from each pair is described as being either 'consistent', 'borderline', or 'unlikely' to have been observed given this hypothesis.'

We have clarified in the figure legend what we mean by being consistent with a process of continuous transmission (please see below); there are multiple potential networks of transmission between the individuals that could be constructed out of consistent pairwise events. Individuals were indeed ordered by date of first positive test.

Figure 3c: A continuous transmission chain of SARS-CoV-2 infections in cluster 1 commenced with a single introduction. Relationships between individuals in cluster 1 were calculated within A2B-COVID. Colours denote potential transmission events from the donor (vertical axis) to the recipient (horizontal axis) that are consistent with transmission¹² or which are borderline possibilities (yellow). The plot shows that the data are consistent with a continuous transmission chain of SARS-CoV-2 infections in cluster 1 occurring via a single introduction; there are multiple potential networks of transmission events between these individuals for which each event would be consistent with a statistical model of direct transmission. We note that individuals in this plot are ordered by the date of the first positive COVID test.

Reviewer 2

In “Genomic epidemiology of SARS-CoV-2 in a UK university identifies dynamics of transmission” the authors provide a descriptive study of how SARS-CoV-2 transmitted within a University setting. This setting of high sampling coupled with sequencing, contact tracing and epidemiological investigations provides a mostly controlled environment to study transmission dynamics. The piece is well written and provides valuable insights into the risk factor and mitigation strategies relevant for this younger population who, across the world’ will likely be in the latest tranches of vaccination and therefore most reliant on non-pharmaceutical interventions for longer.

I would recommend this for publication after minor revisions as I do feel there are several areas in which the paper could be improved to maximise the impact and clarify some of the findings for a wider audience.

We thank Reviewer 2 for this concise summary of the manuscript, and thank the reviewer for highlighting the importance of this study. We hope to have addressed the minor revisions suggested adequately.

For the wide readership of this journal it might be beneficial to include a transmission network visualisation of especially cluster 1 to visualise the repeated ‘mass spreading’ at the social event and spread (or lack thereof) within households. I.e. all current figures are focused mostly on cases, in such a high sampling setting it would be helpful to visualise how many individuals did not test at all vs. did not test positive/were not infected.

We thank Reviewer 2 for this comment. We have now visualised the initial outbreak linked to Venue A in Supplementary Figure 5; here directionality can be more reliably inferred from the epidemiological information and reinforces the important underlying reason for continued spread of Cluster 1: dispersion into multiple households. The paper currently focuses on the genomic epidemiology of positive cases which have been sequenced. Networks are currently inferred through phylogenetics with epidemiological associations then evaluated to understand the underlying transmission dynamics within the university. A network visualisation of cluster 1 representing all infections that were seen has been shown phylogenetically, including associations with the social event. Visualising all individuals in households that were tested vs not tested/not infected would be impractical to plot (because of the large sample size) or at least highly complex, with little additional information gleaned from what is already presented in the manuscript. Further, we quantify household attack rates to provide an understanding of infection rates by using genomic information to refine existing methodology. Additionally, information such as success rate of the screening programme is presented in Warne et al. 2021 Research Square.

The statistical test performed on the ‘SNP differences between cases’ does not account for the fact that the University samples were geographically and socially more closely related, comparing these to ‘randomly’ sampled community cases will distort the importance of this observed lack of

phylogenetic diversity. If a sufficient sample size is available from a smaller region/sub-region with greater social cohesion that is comparable to the University this would be preferable. If the statement of significance is intended to show genomic similarity as a reflection of 'interaction'/cohesion the statement could be caveat-ed as such to ensure the reader is aware of the underlying population structure differences.

We thank Reviewer 2 for raising this important point. We agree comparisons of SNP differences between the university and community do not currently outline the fact this is likely given greater social cohesion within the university. We have therefore clarified this in the caption of Supplementary Figure 4:

“Supplementary Figure 4: The SNP difference among university students was much lower (Wilcoxon signed-rank test, p -value $< 2.2e-16$) than among the rest of the Cambridgeshire community. This is likely to reflect the fact samples from University students were geographically and socially more closely related, and the establishment of fewer persistently transmitting lineages.”

Line 167-169 and 342-344, while the COG-UK consortium has done a truly momentous amount of work the fact that community genome coverage is usually between 8-12% means that statements like “suggesting that the university cases were introduced from outside Cambridgeshire” would be best caveat-ed with a (Bayesian) probability and confidence interval to illustrate the uncertainty in the data.

We agree with the reviewer that it is impossible to exclude the possibility that university cases in cluster 1 were introduced from a previously circulating transmission chain within Cambridgeshire, and have now amended the manuscript to include this caveat (lines 397-398). However, we feel that adding a probability to this possibility without making very broad assumptions about the distribution of lineages and the population structure is complicated and likely beyond the scope of this study. We note that this lineage (B.1.160.7) was not observed in the UK before Oct. 4 (in Wales) and shortly thereafter in the University (Oct. 8). It was not observed in the local community before week 3 of term (Oct 19-25), and in contrast to the university population, only in very small numbers (Fig 3b). Prior to Oct. 2020 case numbers in the UK were low and it is unlikely the lineage would have escaped detection for long. While it is possible that the lineage was first introduced into Cambridgeshire and very shortly thereafter into the university population, the earlier observation amongst university students and the influx of students from across the country (and outside of the UK) at the start of term leads us to conclude that this is the less likely chain of events.

Line 256-259, this statement reads like there were no shared courses/areas/etc.. identified in 7 out of the 8 genomics clusters. What does this mean for the perceived completeness of the data set on both the genomics and epidemiological front? Some wording to explore these cases of missing data vs missing links might be useful. Line 355-358 does not seem to reflect on this finding in the current version.

We agree with the reviewer's statement and have added a sentence in the limitations to reflect this (lines 398-401):

'Further, epidemiological links are dependent on self-reporting and therefore some data will be missing; whilst a lack of epidemiological association between groups in clusters is important and reassuring (such as between staff and students), it does not confirm a lack of transmission.'

There seems to be limited information provided on what statistical/network/other analysis were performed on the epidemiological data and how these could be replicated on other datasets. Would it be possible to share the de-identified contact matrices and household/course data with associated code to enable others to re-create Figure 3c and other epidemiologically informed analysis?

We thank Reviewer 2 for raising this important point. Viral genome sequences and limited associated metadata are made publicly available on GISAID, uploaded to MRC-CLIMB, and COG-UK (<https://www.cogconsortium.uk/tools-analysis/public-data-analysis-2/>) in line with COG-UK ethical approval. These include:

- All sequences
- Trimmed and masked alignment
- Unmasked alignment
- Metadata:
 - sequence name
 - sample location (at a national level)
 - sample date
 - epi week of sampling
 - Pango-lineage (a dynamic variable)
 - mutations of interest
- Tree (Newick format) of GISAID and COG-UK samples

Further, limited public metadata, code, and analysis files for this work are available from GitHub at <https://github.com/COG-UK/camb-uni-phylo>, which also contains a list of ENA study sequence accession numbers. Similarly, limited metadata regarding contact tracing and other epidemiological information can be made available from university students; students are provided with the following privacy notice to encourage enrolment to SARS-CoV-2 screening:

'Monitoring, evaluation and scientific research: Personal data collected about the programme may be processed to support monitoring and evaluation of this programme, and for research in the public interest by the University of Cambridge and other researchers. Wherever possible, this will be done using pseudonymised information (i.e. information with your identifying details removed and not passed on). If any data is presented or published, the data will be fully anonymous, without any means of identifying you. Researchers will abide by codes of ethical conduct.'

Further, descriptive contact tracing work has been conducted with Public Health England as part of surveillance for COVID-19 infections under the auspices of Section 251 of the NHS Act 2006 and/or

Regulation 3 of The Health Service (Control of Patient Information) Regulations 2002. Requests for individual level data would require a direct request to the public health agency.

Specifically, Figure 3c represents an A2B-COVID transmission matrix. A2B-COVID is publicly available on GitHub (<https://github.com/chjackson/a2bcovid>) to allow other researchers to replicate findings on other datasets. Figure 3c was based solely on dates of infection and the viral genome sequences (available as outlined above). Household and course links were determined through SNP matrices and integrating epidemiological data with transmission clusters identified with the CIVET tool (outlined in methods, lines 481-483, 491-493, and 558-560).

Qualifying what data is considered 'basic' (line 575) would be helpful for people to know where they should go to find/obtain certain data

As outlined above we have made as much data as possible available without the risk of deductive disclosure of the individuals involved or breaching the permission granted by the participants of the university screening program. Requests for data relating to the contact tracing would need to be made to the Public Health England office of data release.

It is interesting to see that B.1.1.7, which is considered to spread more easily/rapidly, did not lead to increasing case numbers within the college while it did in the community at the same time. Was there anything particular about the B.1.1.7 cases in the University setting that prevent spread?

We thank Reviewer 2 for highlighting this interesting point. Indeed, B.1.1.7 did not lead to increasing cases in the university despite being associated with increased transmissibility relative to circulating lineages at the time.

Two infections of Pango-lineage B.1.1.7 were seen amongst the university students. Both infections occurred during the national lockdown but were not related epidemiologically and did not overlap temporally, nor were there any links with the community. No further infections were seen amongst the infected individuals' households and therefore there was no further opportunity for spread from these two cases. This is a likely reflection of the fact that these infections occurred during the national lockdown at a time where the student population were highly sensitised to the infection prevention control measures in place; a demonstration of how such measures can be effective against even highly transmissible variants. Further, this also highlights the over-dispersed nature of SARS-CoV-2 transmission chains and the role played by stochasticity in its spread. We have outlined these findings in the results for readers (lines 147-149).

Figure 2 a/c, it might be more intuitive to have all B.1.177.Xs be a different shading of the base lineage colour to reflect the phylodynamic relations between lineages and to retain black or grey for 'other'

We agree this is a more intuitive way of displaying Figure 2 a/c. We have therefore adjusted the colours accordingly:

Figure 4 legends of panel B and C appear to be swapped

Thank you for pointing this out. We have now corrected this in the manuscript.

Line 148-154, it would be good to include the estimated/inferred number of introductions into the university to contextualise this statement “signifying that the majority of introductions into UoC did not cause ongoing transmission.” Currently it is unclear if this is ‘5 out of 198’ or a different number.

We thank Reviewer 2 for this comment. We agree this result, presented in this manner, is ambiguous. We have now specifically reported clusters with greater than 5 university cases as a proportion of clusters that contained university students (line 152).

Line 136, it would be useful to know what date/version of Pango this was in text as version changes can have a major impact on the number of lineages possibly identified

Thank you for spotting this omission. We have now added the version of Pangolin used to the methods (line 497).

Line 318: “demonstrating both compliance and an effective control strategy.” Is it not one leading to the other?

We thank Reviewer 2 for pointing this out. We have changed the text to clarify the sentence (lines 366-367):

“demonstrating high levels of compliance from our study population with an effective control strategy”.

Methods; when referring to a specific version it is useful to refer to the specific tag like this: <https://github.com/tseemann/snp-dists/releases/tag/v0.7.0>.

Thank you. We have adjusted the text to refer to specific software versions with the tag as suggested, where possible.

In addition, some of the links in the methods section do not seem to be properly hyperlinked.

Thank you. We have adjusted the text to ensure all links are properly hyperlinked.

In the legends of supplement figures, specifically the CIVET + microreact trees, there is no specification of what the square symbol represents.

We thank Reviewer 2 for this comment. Sequences from GISAID used to contextualise study sequences by CIVET are represented by dark grey squares (collapsed nodes) and dark grey circles (individual sequences). We have adjusted the caption of Supplementary Figures 9 and 11 to provide an explanation of the figures generated with CIVET.

Reviewer 3

This manuscript presents a detailed, well written, polished outbreak investigation focused on The U of Cambridge between 5 October and 6 December 2020. However, while the paper is well written, the conclusions don't directly flow from the information as it has been presented. While the focus is on the university setting, there is only sporadic, broad information about the specific university COVID-19 control measures in the discussion and supplementary information, and the authors have not really drawn together specific information on the timing and detail of control measures at the university level, and the changes in case numbers/Re, in order to make any suggestion of impact. This would have made the paper much more compelling. I suspect that there was not a great deal of difference in the local university control measures over time, and the real change was the implementation of the national lockdown, which is perhaps why this has been glossed over.

We thank Reviewer 3 for this comment. This paper aims to describe the factors which resulted in extensive outbreaks within the university and factors which led to control. We highlight how confining outbreaks to accommodation settings, though a high risk setting for transmission in itself, can result in cessation of outbreaks. We demonstrate, in a large cluster, how asymptomatic testing helped identify (and isolate) cases which would have been otherwise missed. Further, the screening programme ran throughout the 1st university term and thus we report inferred statistics such as the effective infectious period and household attack rates to display further efficacy of the total measures used in the university, with the screening programme the most significant. We agree that the paper could benefit from further clarity and detail. We have now provided a more detailed

combined evaluation of university measures and observations within the manuscript. Further, we have now also provided extensive additional information detailing university-wide measures to mitigate SARS-CoV-2 transmission in the Supplementary Materials.

While the manuscript provides a comprehensive, polished genomic epidemiological examination of an outbreak in a university setting which is certainly of local interest, it is difficult to discern the broader implications of the findings presented. Transmission in household, workplace and social settings, including nightclubs, has been well described previously. Without further clarification of what is really unique about this report it is hard to justify publication in this journal.

We thank Reviewer 3 for highlighting that the work is 'comprehensive and polished'. To our knowledge, there is still no detailed report on the genomic epidemiology of SARS-CoV-2 outbreaks within a university or other higher education settings; interactions in households, the workplace, and social settings outside of a university would have involve populations from a different demographic, be of smaller numbers, and reflect different types of social interactions (i.e. duration and nature). We do believe this work is of global interest; the young population in higher education settings remain largely unvaccinated across the globe as, Reviewer 1 points out, and this work will help direct IPC measures in these settings. Further, this work provides a detailed piece through dense sampling of the university members and community prior to the establishment of a vaccination programme or the emergence of variants of concern – this unique dataset will be vitally important to compare future studies to. Finally, the comparison with the community, through the extensive community sampling by COG-UK, provides a unique report on the interaction of university members with its community, a source of significant concern (<https://www.gov.uk/government/publications/principles-for-managing-sars-cov-2-transmission-associated-with-higher-education-3-september-2020>). We thank Reviewer 3 for highlighting the need for additional clarification of what makes this study unique in the paper; we have provided additional information reflecting the discussion above in the manuscript introduction (lines 107-120) and discussion (lines 303-311). Further, we have provided a more thorough discussion tying together Infection Prevention Control measures within the university and findings in our study to assist other higher education facilities to mitigate SARS-CoV-2 transmission.

Abstract, line 75-76. The author's note that the study highlights effective interventions in a higher education setting, but these have not been well described or explored in the manuscript. Suggest authors considering including the extent and, particularly, timing of UoC and national interventions in the main manuscript and how these relate to the observed changes in case numbers and Re. Without this, the paper is not highlighting any direct interventions that could be used to more safely open up universities in future.

We are grateful to the reviewer for highlighting this issue with the paper. As described above, we have made extensive changes to the supplementary materials and the discussion to more closely align the genomic and epidemiological findings with the interventions introduced into the University over the study period.

One of the key determinants of the success of the interventions was that they were multidisciplinary, bespoke to the individual situation and use multiple approaches to reduce

transmission. It has therefore not been possible to describe in detail all of the interventions introduced for each cluster. However, we have now provided a more detailed description of the University's key policies for infection control over the autumn term, as well as a more comprehensive narrative of the control of one example outbreak, referred to in the manuscript as cluster two. The discussion of cluster two in the supplementary materials is detailed as follows:

'For example, one of the first clusters of cases to be identified within a single block of accommodation in one college and managed as an outbreak was detected during the second week of term. The first individuals to be identified were screened via the asymptomatic screening pathway. Over the following 2 days further students were identified through the university's symptomatic testing route. All suspected and confirmed cases, and their households, were immediately isolated and contact tracing initiated as described above. Subsequent genomic analysis confirmed that the majority of these cases were linked isolates, and are described in the results as cluster two. Within four days of the index case testing positive an extraordinary meeting was held between members of the college, university and the local public health authority who agreed an immediate lockdown of the affected accommodation block in its entirety. Students were supported with deliveries of food and drink, their educational needs were discussed individually between students and their tutors, and additional psychological support was provided as necessary. However, students were not allowed to leave the accommodation block unless they were attending an appointment for SARS-CoV-2 testing or another valid medical reason. In addition to the existing availability of symptomatic testing from the university, individual asymptomatic screening was offered to all students living in the accommodation block over the following four days. These measures were successful at reducing the number of cases within both the accommodation block and the wider college. As described in Results, subsequent genomic analysis has demonstrated this viral lineage became extinct in the study population within two weeks of the accommodation block being placed under isolation.'

We have twice identified the implementation of lockdown, either in a local setting in the case of cluster two, or a broader setting in the implementation of national lockdown, to be highly effective at reducing transmission in our setting. We have provided additional information on the measures implemented as part of the University's routine COVID-19 reduction measures and its specific response to transmission events within colleges in the supplementary materials.

Discussion, lines 314-315. The author's note a rise in reproductive rate following the announcement of national lockdown, and suggest the need for a targeted public health campaign to reduce high risk activities during the 'lag' period. Can the authors justify why they have not suggested that the time between announcement and implementation of the lockdown be reduced? In some settings such actions are implemented ~24 hours following announcement, ensuring interventions are implemented sooner and reducing opportunity for geographical dispersion and increase engagement in high-risk behaviour during this period.

We thank Reviewer 3 for this comment and completely agree reducing time to announcement of national lockdown should be a consideration given our findings. We have adjusted text accordingly (lines 358-360):

'There was a rise in the effective reproduction number coinciding with the announcement of a national lockdown on 31 October, to begin on 5 November 2020. This announcement prior to implementation of a major socially restrictive public health measure, alongside existing Halloween festivities, may have led to increased levels of behaviour associated with higher risk of transmission. This supports either reducing the time from announcement to implementation of socially restrictive measures, or the need for a targeted public health campaign to limit high risk activities where this is not possible.'

Figure 2c: Can the authors suggest why a steeper decline in SARS-CoV-2 cases was observed within UoC than in the local community, and how the interventions in place at the time of the observed decrease differed between settings?

This is challenging to answer because the adherence to control measures in the community was not measured. However, we know from contemporary studies elsewhere in the UK that adherence to COVID-19 prevention measures is mixed, especially among young adults (reference - <https://www.bmj.com/content/372/bmj.n608>). This analysis of 37 cross sectional surveys indicate that although young age is a risk factor for poor adherence, other key determinants relate to the penalties that individuals experience during lockdown (such as having a dependent child in the household, financial hardship and working in a key sector), but these are much less common concerns within the university population. Although no direct incentives were provided to individuals, the expectations of individuals to adhere to rules was communicated widely. We also believe a key to the successful implementation of lockdown measures was the additional support provided by the collegiate university, ranging from practical provision of food and drink through to the pastoral and community support provided by established networks of staff, tutors and student representatives.

We are grateful to the reviewer for raising this point and have amended our discussion to reflect this (lines 365-377).

Conclusion, lines 366-367: The authors conclude that 'the insights gained will inform public policy regarding infection control measures in higher education settings', however limited attention has been paid to the timing and impact of local (and national) control measures. If the author's wish to suggest this as a major conclusion of the paper, I suggest a closer examination of the detail and timing of specific control measures and the changes in Re/case numbers is provided.

Please see the response above.

Discussion line 351: The authors advocate for a combined genomic epidemiological approach to outbreak investigations (and presumably surveillance more broadly). Can the authors please comment on if genomic data was used prospectively to inform control measures throughout the study period, or if all genomic analyses were performed retrospectively?

We thank Reviewer 3 for this insightful comment. This study was designed to be retrospective and preliminary findings were presented in an interim report to the UK government Scientific Advisory Group for Emergencies (SAGE) in December 2020 and to the regional Covid Incidence Management Team on 23rd Nov. At this stage however, the UK was placed under national lockdown and this was the most effective infection prevention control measure. Our findings did provide reassurance that

mass screening and sequencing was useful for a) asymptomatic case detection and b) outbreak investigation whilst providing important reassurance that little university-community transmission was occurring, informing national policy.

In line with real-time analyses conducted in hospitals (Hamilton et al. 2020 eLife) and community settings (Seemann et al. 2020 Nat Commun), we strongly feel a combined genomic epidemiological approach would be beneficial in mitigating transmission as informed by our detailed analysis.

Additional work is required for the modelling aspects of the paper:

First, the authors used ModelFinder to identify the GTR+Gamma model as the most optimal model for the data, yet they chose to use an HKY+Gamma model in their BEAST analyses. The HKY model does not allow for distinct Transition and Transversion rates, which we know are a feature of SARS-CoV-2 data. They need to redo their analyses with a better model. If HKY is more adequate for the BEAST modelling, then this should be outlined clearly for the reader.

Our substitution model choice for the BEAST analyses was informed by previous SARS-CoV-2 analyses (Duchene et al. 2020 Viral Evolution, Nadeau et al. 2021 PNAS, Vaughan et al. 2020 medrxiv, Hodcroft et al. 2021 Nature, Geoghegan et al. 2020 Nat Commun). We do however agree with the reviewer that using the same substitution model throughout is the more elegant choice and that a GTR+Gamma model could help refine estimates as it is more flexible. We have therefore re-run both the coalescent and BDSKY analyses using a GTR+Gamma model. We note that estimates of parameters of interest are unchanged, showing that our conclusions are robust to changes in the substitution model.

Second, the choice of fixing the substitution rate to some generic value from Rambaut et al. 2000 seems unwarranted. There are good estimates for the substitution rate from other papers they could use, for instance Hoshino et al. 2021 Gene, Pipes et al. 2021 MBE and Duchene et al. 2020 Viral Evolution, Seemann et al. 2020 Nat Commun. In addition, I would suggest including a bigger period of sequence data from the UK and the rest of the world to get a more robust temporal signal, thus precluding the need to set such a heavy prior on the substitution rate — one that will affect the time estimates of internal nodes and all other derived estimates.

We thank Reviewer 3 for highlighting more recent estimates for the substitution rate. We have adjusted our substitution rate selection accordingly, choosing $8e-4$ s/s/y in line with analyses by Ghafari et al. 2021, medrxiv, Vaughan et al. 2020, medrxiv, Nadeau et al. 2021 PNAS, Hodcroft et al. 2021 Nature, and Geoghegan et al. 2020 Nat Commun. We note that Duchene et al. 2020 Viral Evolution and Seemann et al. 2020 Nat Commun suggest a substitution rate of $1.1e-3$ s/s/y. Since our dataset is more recent and sampled over a longer time period we would expect a slower clock rate (in line with the references above). In addition to the main analyses using a clock rate of $8e-4$ s/s/y we now also provide the output of the coalescent and BDSKY analyses using a faster clock rate of $1e-3$ s/s/y (and a GTR+gamma substitution model) in Supplementary figures 7 and 10, showing that our conclusions are robust to this change in the clock rate.

Finally, I think using a strict clock is probably unwarranted, and a relaxed clock is likely to fit the data better. It is good practice to examine different priors on the clock by running models with distinct priors and comparing the models. The best models according to the data can then be used to obtain final estimates of interest (I suggest Stadler et al. 2013 PNAS as a good example to follow). The same applies for the BDSK models.

We thank Reviewer 3 for this insightful comment. The strict clock is a good fit for SARS-CoV-2 as the time scale of sampling is short and the rate of sequence evolution not as high as for flu. Nevertheless, we repeated our analyses using a relaxed clock. We observed a lack of convergence in the time-scaled coalescent tree; evidence that the data can't inform that model well. For the BDSKY analyses a relaxed clock resulted in qualitatively similar results, however with slower convergence. We further note that the 95% HPD interval of the coefficient of variation of the clock rate did not exclude 0. This means there is little variation in the estimated clock rate among tree branches, leading us to conclude that a strict clock is the better fitting model. Finally, we also note that a strict clock has been the standard in most SARS-CoV-2 papers so far (Nadeau et al. 2021 PNAS, Hodcroft et al. 2021 Nature, Du Plessis et al. 2021 Science, Tegally et al. 2021 Nature Med, Kraemer et al. 2021 Science, Geoghegan et al. 2020 Nat Commun).

REVIEWERS' COMMENTS

Reviewer #2 (Remarks to the Author):

I thank the authors for their thorough responses to the suggestions raised by myself and the other reviewers. The proposed revisions address the comments well. I would still encourage the authors to think about how to better document the procedure researchers could follow to obtain the more detailed epidemiological data. We have seen solutions like EGA for human genome information, it would be good to see what such a system could look like for epidemiological data. For this paper a reference to PHE with contact information would be a good compromise. I am happy for this manuscript to be published in its revised form.

Reviewer #3 (Remarks to the Author):

Thank you for the opportunity to review this revised manuscript. The authors have comprehensively addressed my previous comments, and present a manuscript that will be a valuable addition to the literature.

The addition of details regarding mitigation strategies at UofC and clarification regarding the value add of this study are valuable.

I have no further suggested changes.